# Representation of visual uncertainty through neural gain variability

Olivier J. Hénaff [1,4], Zoe M. Boundy-Singer[2], Kristof Meding[3], Corey M. Ziemba [2] & Robbe L. T. Goris[2✉]

Uncertainty is intrinsic to perception. Neural circuits which process sensory information must therefore also represent the reliability of this information. How they do so is a topic of debate. We propose a model of visual cortex in which average neural response strength encodes stimulus features, while cross-neuron variability in response gain encodes the uncertainty of these features. To test this model, we studied spiking activity of neurons in macaque V1 and V2 elicited by repeated presentations of stimuli whose uncertainty was manipulated in distinct ways. We show that gain variability of individual neurons is tuned to stimulus uncertainty, that this tuning is specific to the features encoded by these neurons and largely invariant to the source of uncertainty. We demonstrate that this behavior naturally arises from known gain-control mechanisms, and illustrate how downstream circuits can jointly decode stimulus features and their uncertainty from sensory population activity.

[1] Center for Neural Science, New York University, New York, NY, USA. [2] Center for Perceptual Systems, University of Texas at Austin, Austin, TX, USA. [3] Neural Information Processing Group, University of Tübingen, Tübingen, Germany. [4] Present address: DeepMind, London, UK. ✉email: robbe.goris@utexas.edu

Sensory systems offer a window onto a world that cannot be known perfectly. Uncertainty about the world can arise externally, when sensory cues are incomplete or contradictory, or internally, when noise corrupts neural representations. Ideal perceptual systems take this uncertainty into account: if a sensory cue is ambiguous, prior experience guides its interpretation[1], and when multiple cues are available, they are combined in proportion to their reliability[2]. When humans and other animals perform perceptual tasks, they often follow these normative predictions[3–6].

These behavioral effects imply that the neural circuits which mediate perception assess the uncertainty of sensory information. How they do so is unclear. A prominent hypothesis is that the same neurons that encode a stimulus feature also encode the uncertainty about this feature[7–9]. However, which aspect of neural activity represents uncertainty remains a topic of debate. It has been argued that response variability is a promising candidate[9]. In visual cortex, it is maximal in the absence of a stimulus[10] and declines with contrast[11], aperture size[12], and attention[13,14]. Since each of these factors is associated with increased information about the visual environment, response variability might represent stimulus uncertainty.

Here, we incorporate this hypothesis into the canonical model of neural coding. We propose that, while average response magnitude encodes stimulus features, variability in response gain encodes the uncertainty of these features. We formalize this proposal in a doubly stochastic response model in which spikes arise from a Poisson process whose rate is the product of a deterministic response mean and a stochastic response gain. The mean response is governed by a parametric function commonly referred to as the classical receptive field. We introduce a second function, the uncertainty receptive field, which determines the variance of the response gain.

To test our theory, we studied responses of individual orientation-selective neurons in macaque visual cortex, driven by repeated presentations of stimuli whose orientation uncertainty was manipulated in two different ways. As predicted, we found that gain variability selectively depends on stimulus uncertainty, and that this selectivity is roughly invariant to the source of uncertainty. This appears to be a general property of visual coding: we find that the gain variability of texture-selective neurons in V2 systematically increases with an image's textural uncertainty. To identify the neural computation that gives rise to this behavior, we developed a probabilistic model of divisive normalization in which driving input is divided by noisy suppressive inputs. This model quantitatively matches the effects of stimulus uncertainty on response variability.

Finally, we asked whether our coding scheme permits downstream circuits to quickly decode the information needed for perceptual tasks. We find that neuronal gain exhibits slow dynamics, not fast. Consequently, gain variability cannot be readily decoded from individual neurons. We derived an optimal decoder of neural population activity, and used model simulations to investigate its performance. We show that stimulus orientation and gain variability can be jointly decoded from a brief V1 population response and that gain variability faithfully predicts the accuracy of orientation decoding. Together, these results establish cross-neuron variability in response gain as a candidate currency of uncertainty in sensory cortex.

## Results

### Expanding the canonical model of neural coding. 
In primary visual cortex (V1), neurons are tuned for local image orientation, making this area well suited to inform perceptual orientation estimates. An effective estimation strategy is to consider the

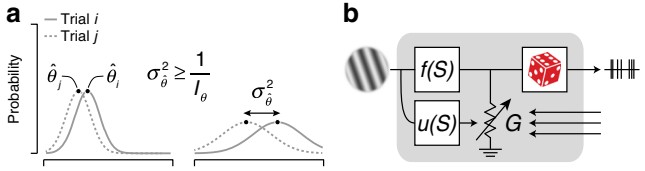

**Fig. 1 Encoding information for perceptual tasks. a** An effective strategy for estimating a perceptual quantity such as local image orientation is to compute the likelihood of each possible stimulus value (gray lines) and choose the most likely option ($\hat{\theta}$, black dots). Due to noise, this value will differ across trials (full vs dotted lines, $\hat{\theta}_i$ vs $\hat{\theta}_j$). The larger the uncertainty of the sensory information, the wider the likelihood function, and the greater the variance in the perceptual estimate (left vs right panel). The variance of this maximum-likelihood estimate $\sigma_{\hat{\theta}}^2$ is larger than or equal to the inverse of the Fisher information $I_\theta$[20]. **b** Schematic summarizing the proposed model. Spikes arise from a Poisson process whose rate is the product of a deterministic drive $f(S)$ and a stochastic gain $G$. $f(\cdot)$ governs the mapping of stimulus features onto drive and hence controls response mean; $u(\cdot)$ governs the mapping of stimulus uncertainty onto gain variability and hence controls response variance.

probability of each possible orientation given the V1 population response, and select the value that is most likely. However, because of internal and external noise, this likelihood function and the resulting orientation estimates vary from trial to trial (Fig. 1a, left). The lower the signal-to-noise ratio, the greater the uncertainty, and the greater the variance of the estimate (Fig. 1a, right).

Many perceptual tasks require that the uncertainty of perceptual estimates be assessed on a moment-by-moment basis. How can downstream circuits instantaneously assess the reliability of V1 orientation reports? Since this reliability varies systematically with certain features of the stimulus such as the size and contrast of a local image patch, V1 neurons might encode reliability through a separate channel tuned to these features[9]. Specifically, let us assume that a neuron's response is in part governed by a deterministic function of the stimulus $f(S)$ (the classical receptive field) and in part by noise (Fig. 1b, top branch). Previous work has shown that spike counts $K$ are well described by a modulated Poisson process whose rate is the product of $f(S)$ and a stochastic response gain $G$[15]. In particular, if the gain $G$ has a unit mean and varies on a time-scale which is slow relative to the measurement interval $\Delta t$, spike-count variance can be decomposed as

$$\mathrm{Var}[K|S, \Delta t] = f(S)\Delta t + \sigma_G^2(f(S)\Delta t)^2. \tag{1}$$

The first term is the variance due to the Poisson process, the second is due to variability in the firing rate and grows with the variance of the gain $\sigma_G^2$. Whereas this gain variance was originally assumed to be stimulus independent[15], we propose that it systematically depends on the stimulus via an uncertainty receptive field $u(S)$ (Fig. 1b, bottom branch). If the uncertainty receptive field is selective for stimulus features that induce uncertainty, gain variability may provide a useful assay for the reliability of V1 orientation reports.

The classical receptive field is associated with two key properties: it endows sensory neurons with a particular selectivity and a particular invariance. For example, the firing rate of V1 complex cells reports the total amount of energy in a particular orientation range, irrespective of the image's polarity or precise location within the receptive field[16]. We hypothesize that the computations underlying the uncertainty receptive field achieve a similar effect. Specifically, we expect that the gain variability of sensory neurons reports the total amount of uncertainty about the

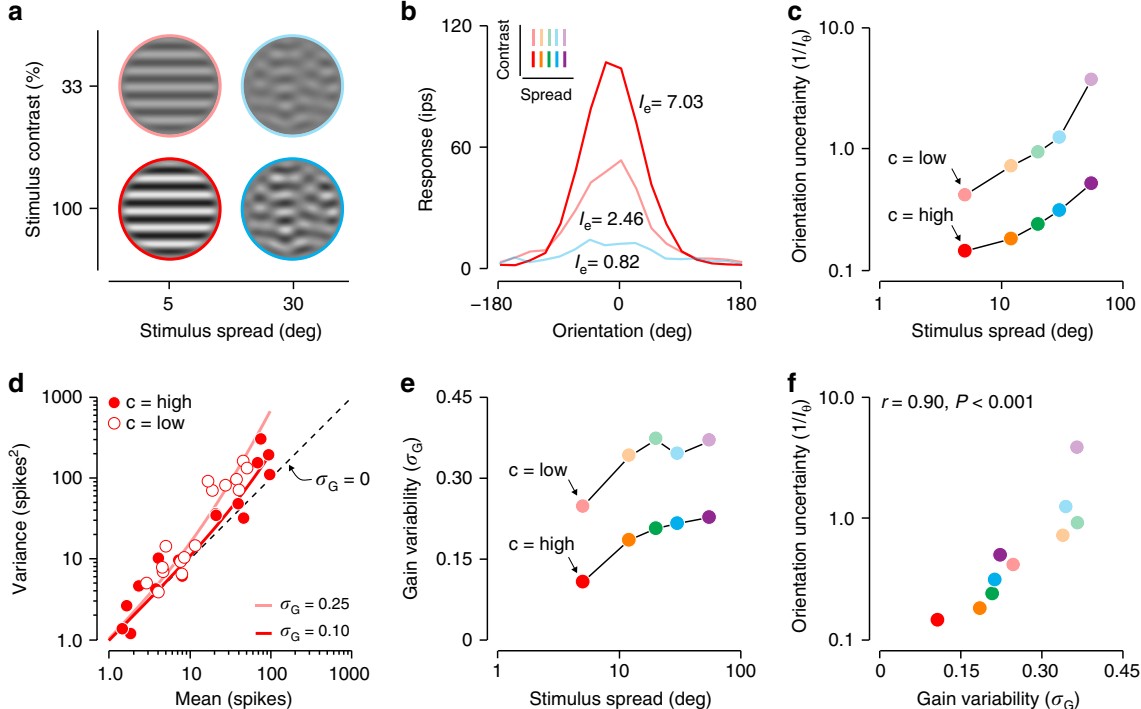

**Fig. 2 Estimating stimulus uncertainty and gain variability. a** Each stimulus consisted of a sum of drifting sinusoidal gratings, with drift directions drawn from a Gaussian distribution. Stimuli differed in center drift direction (16 levels), spread (5 levels, represented by hue), and contrast (2 levels, represented by saturation). **b** Mean responses of a V1 neuron as a function of drift direction for three stimulus families. Responses were computed by counting spikes in a 1000 ms window following response onset. From these tuning curves, we estimated Fisher Information ($I_\theta$, see online Methods). We use its inverse as a measure of orientation uncertainty. **c** Orientation uncertainty as a function of stimulus contrast and spread for the example neuron. **d** Variance-to-mean relation of the example neuron for the narrowband stimuli. Different points indicate different drift directions. Lines illustrate the predictions of the modulated Poisson model, fit separately to the high- and low-contrast conditions. **e** Gain variability as a function of stimulus contrast and spread for the example neuron. **f** Orientation uncertainty as a function of gain variability for the example neuron ($r = 0.90$, $P < 0.001$).

features they represent, while being invariant to the source of this uncertainty.

**Testing the theory in visual cortex.** To test our theory, we analyzed responses of neurons in macaque visual cortex elicited by mixtures of sinusoidal gratings (Fig. 2a; a model-based analysis of these data concerned with mechanisms of orientation selectivity has been previously published[17]). These stimuli are Gaussian-distributed in the orientation domain, hence the perceptual uncertainty about their orientation depends on only two factors: the total amount of stimulus energy (contrast), and its dispersion (spread). Indeed, increasing stimulus spread increases perceptual discrimination thresholds because it acts as external orientation noise[18]. Reducing stimulus contrast has the same effect because it exposes internal noise[19].

These behavioral effects are mirrored by changes in coding capacity at the level of individual neurons. Consider the orientation information encoded in the response of an example neuron to a narrowband stimulus. Reducing stimulus contrast from 100 to 33% approximately halved this neuron's mean response (Fig. 2b). To determine the impact of this loss of responsivity, we estimated the Fisher information associated with both conditions ($I_\theta$, see Online Methods). This statistic quantifies the amount of orientation information that can be extracted from the neuron's responses by an optimal decoder. Specifically, its inverse provides a lower bound on the variance of the maximum-likelihood estimate[20], and we use it here as a proxy for orientation uncertainty. For the high-contrast stimulus, the Fisher information was 7.03; for the low-contrast stimulus, it was 2.46 (Fig. 2b).

For this neuron, the contrast reduction thus led to a substantial increase in orientation uncertainty. Increasing stimulus spread had the same effect (Fig. 2b), which was evident both at high and low contrast (Fig. 2c).

Are these changes in stimulus uncertainty reflected in the neuron's gain variability? We used the modulated Poisson model to estimate gain variability for each stimulus family separately (Online Methods). For the narrowband stimulus, gain variability was greater at low contrast than at high contrast (Fig. 2d; $\sigma_G = 0.10$ at high contrast, $\sigma_G = 0.25$ at low contrast). Moreover, gain variability also increased with stimulus spread, irrespective of the contrast level (Fig. 2e). Across all stimulus families, orientation uncertainty and gain variability exhibited a striking quantitative relationship ($r = 0.90$, $P < 0.001$; Fig. 2f).

The dependency of gain variability on stimulus uncertainty was evident across the population of V1 and V2 neurons. There was some heterogeneity in the effects of the stimulus manipulations on neurons' responses[17], but overall, both manipulations substantially increased orientation uncertainty (stimulus contrast: $P < 0.001$, $F_{1,783} = 48.18$, ANCOVA; stimulus spread: $P < 0.001$, $F_{1,783} = 188.72$). This can be clearly seen in the stimulus uncertainty estimates, averaged across neurons (Fig. 3a). Moreover, the uncertainty manipulations did not interact significantly ($P = 0.86$, $F_{1,783} = 0.03$; Fig. 3a), suggesting that they independently contribute to stimulus uncertainty. The average gain variability was monotonically related to the average uncertainty value (Fig. 3b). This suggests that gain variability may represent the total amount of stimulus uncertainty, regardless of the source of this uncertainty (stimulus contrast: $P < 0.001$, $F_{1,783} = 94.13$, ANCOVA; stimulus spread: $P < 0.001$, $F_{1,783} = 32.58$). Closer

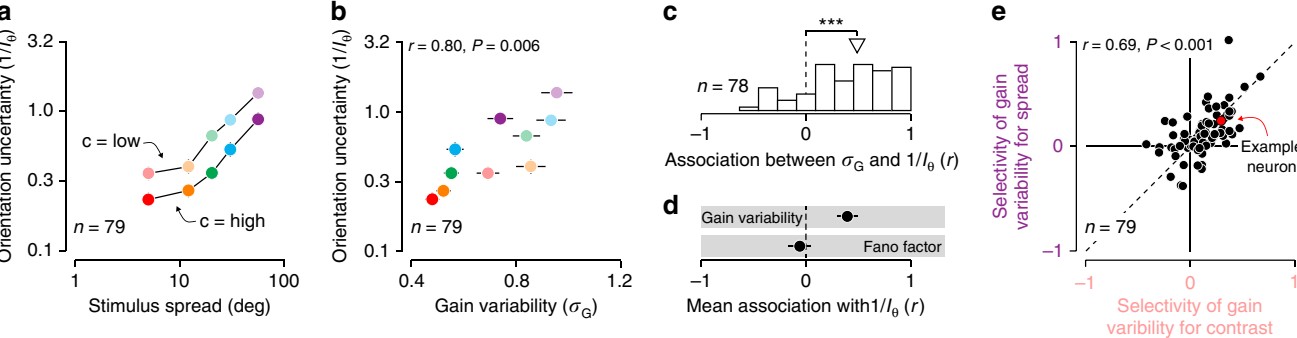

**Fig. 3 Gain variability represents stimulus uncertainty. a** Orientation uncertainty as a function of stimulus contrast and spread, averaged across a population of V1 and V2 neurons. **b** Orientation uncertainty as a function of gain variability, averaged across a population of V1 and V2 neurons ($r = 0.80$, $P = 0.006$). **c** Distribution of the correlation between gain variability and orientation uncertainty for individual neurons. The triangle indicates the median value. **d** Average correlation between orientation uncertainty and two measures of neuronal response variability. Here, error bars indicate 95 percent confidence intervals. **e** Selectivity of gain variability for stimulus spread as a function of selectivity of gain variability for stimulus contrast (see Online Methods). Error bars indicate ± 1 s.e.m; ***$P < 0.001$.

examination of the behavior of individual neurons revealed that for most units, orientation uncertainty and gain variability are positively correlated (median $r = 0.49$, $P < 0.001$, Wilcoxon signed rank test; Fig. 3c).

Are these results unique to gain variability, or are other measures of the dispersion of neuronal responses also indicative of stimulus uncertainty? For each neuron, we compared two different statistics: gain variability and Fano factor (defined as the ratio of the spike-count variance to the mean, see Online Methods). While gain variability was positively associated with uncertainty ($r = 0.40 \pm 0.04$, mean ± s.e.m.; Fig. 3d), Fano factor exhibited no systematic relation with uncertainty ($r = -0.06 \pm 0.05$). Why is this so? The more uncertain stimulus conditions are associated with reduced responsiveness and increased gain variability. Together, these effects can make the Fano factor detached from stimulus uncertainty (Supplementary Fig. 1).

Finally, we asked whether the gain variability of individual neurons is tuned to stimulus uncertainty per se, or to a subset of the stimulus features that induce uncertainty. We singled out the most extreme stimulus manipulations, both of which induced substantial amounts of uncertainty (minimal spread at low contrast and maximal spread at high contrast). Could it be that different subsets of neurons are selective for each of these manipulations? This would question the existence of a monolithic uncertainty receptive field. We summarized each neuron's selectivity for these manipulations by measuring the change in gain variability relative to the baseline condition (minimal spread at high contrast, see Online Methods). This statistic equals one if the stimulus manipulation increases gain variability by a factor of ten, and zero if the stimulus manipulation has no effect on gain variability (negative values indicate a decrease in gain variability). Interneuronal differences in selectivity for both manipulations were highly correlated ($r = 0.69$, $P < 0.001$; Fig. 3e). This approximate invariance to the source of uncertainty suggests that a single mechanism could account for the uncertainty selectivity exhibited by cortical neurons.

**Representation of uncertainty across the visual hierarchy**. We have, thus far, found evidence for our proposed coding scheme in the relationship between orientation uncertainty and the gain variability of orientation-selective neurons. Our model is not limited to orientation coding, but holds that as new features are encoded along the visual hierarchy, so is their associated uncertainty. In area V2, neurons are selective for the features of visual

texture, a property lacking from their V1 inputs[21]. Our framework therefore predicts that the gain variability of V2 cells, but not V1 cells, will depend on uncertainty about stimulus texture. To test this prediction, we analyzed responses of individual neurons in macaque V1 and V2 elicited by a set of naturalistic textures and a set of unstructured noise stimuli (Fig. 4a–c; data collected by ref. [22]). The noise stimuli were devoid of distinctive textural features and hence induce maximal textural uncertainty —just like a uniformly dispersed stimulus would induce maximal orientation uncertainty. As predicted, noise stimuli typically elicited more gain variability than texture stimuli in V2 (median selectivity of gain variability for textural uncertainty in V2 = 0.063, $P < 0.001$; Fig. 4d, e; see Online Methods). Neurons in V1 showed no such effect (median selectivity of gain variability in V1 = 0, $P = 0.31$; Fig. 4e). These effects are specific to gain variability and do not generalize to Fano factor (Fig. 4f). We conclude that, as neurons' mean firing rates become selective for increasingly complex features of the visual environment, so does their gain variability for the associated uncertainty.

**The uncertainty receptive field arises from normalization**. Which neural mechanism is general enough to support the representation of uncertainty across the visual hierarchy? Divisive normalization is a promising candidate for several reasons. First, this computation is implemented by a wide range of sensory and non-sensory circuits[23]. Second, normalization directly controls neural response gain, and hence might also control gain variability. Finally, divisive normalization can be instantiated in image-computable models (i.e., models that can be evaluated on arbitrary images)[17,24,25], making this a broadly testable hypothesis. We derived a stochastic formulation of the standard divisive normalization model (Fig. 5a; a related model was recently proposed in a separate context[26]). The mean response of this model $f(S)$ is approximately equal to the deterministic version of the normalization model:

$$f(S) = \left( \frac{g(S)}{\beta + \sum_j g_j(S)} \right)^p, \qquad (2)$$

where $g(S)$ is some function of the stimulus, $\beta$ is a stimulus-independent constant, and $p$ is a transduction exponent. The stimulus-dependent normalization factor $\sum_j g_j(S)$ reflects the aggregate activity of a large number of nearby neurons. Neural activity is noisy. We therefore make the normalization term subject to additive Gaussian noise with zero mean and variance $\sigma_N^2$. This makes the firing rate subject to stochastic gain

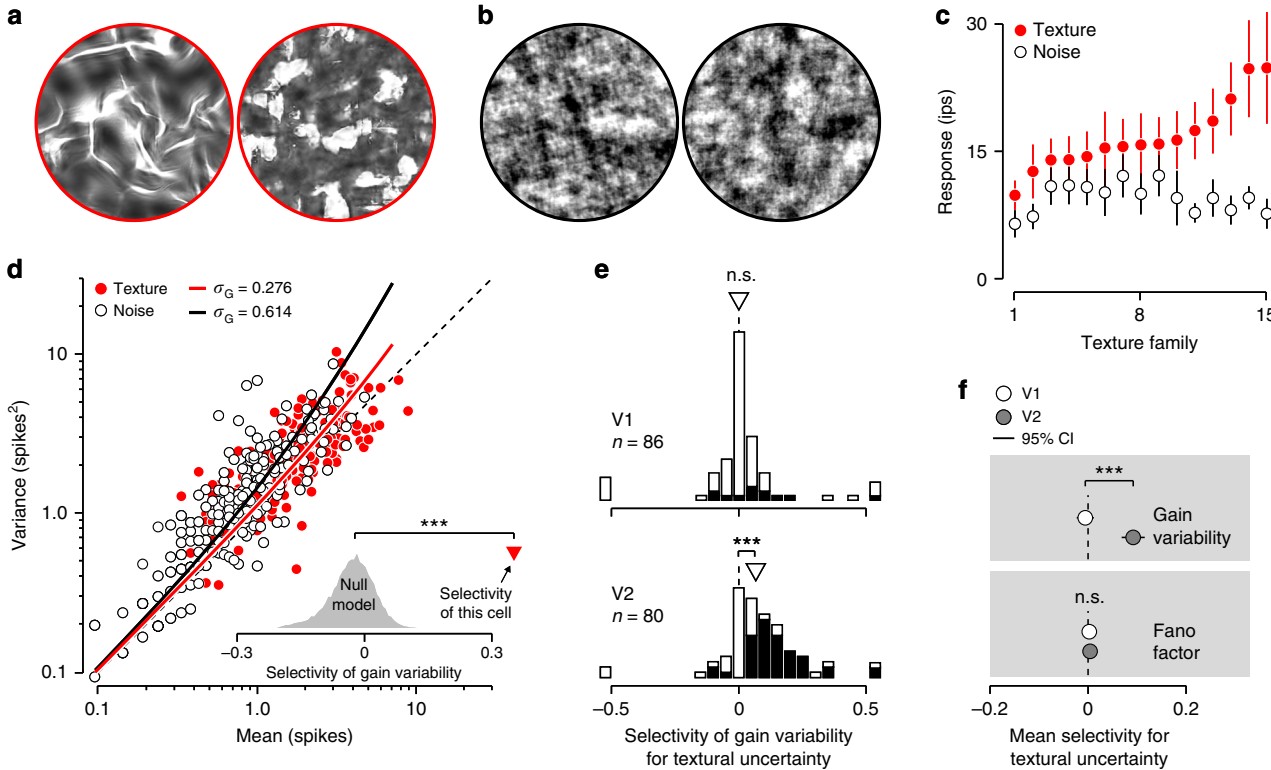

**Fig. 4 Gain variability of V2 neurons represents texture uncertainty. a** Two example images with well defined textural properties (i.e. water, left; flowers, right). **b** Noise images with ill-defined textural properties. Perceptually, the texture images are more distinct than the noise images (C.M Ziemba, J. Freeman, E.P Simoncelli, J.A Movshon, Soc Neurosci. Abstr. 573.13, 2012). **c** This differential perceptual sensitivity is reflected in the stimulus selectivity of V2 neurons, but not of V1 neurons[21]. Mean response of an example V2 cell for 15 families of texture images (red points), and matched noise images (white points). Each family consisted of 15 unique images. Error bars indicate ± 1 s.e.m. **d** Variance-to-mean relation of the same cell for all unique texture and noise images (red vs white points, respectively). Lines illustrate the predictions of the modulated Poisson model, fit separately to the texture and noise stimuli. The inset shows the selectivity of gain variability for textural uncertainty expected under the null model (gray histogram, see Online Methods), and the selectivity estimated from the real data (red triangle). **e** Distribution of selectivity of gain variability for textural uncertainty for a population of V1 (top) and V2 (bottom) neurons. The dotted line illustrates the median value of the null model. Filled bars represent cases that were significantly different from the null model (two-sided test, $\alpha = 0.05$). The triangle indicates the median value. **f** Average selectivity for textural uncertainty computed for two measures of neuronal response variability. Error bars indicate ± 2 s.e.m; n.s. not significant; ***$P < 0.001$.

fluctuations, and yields a simple approximate expression for gain variability (see Online Methods):

$$\sigma_G = \frac{\sigma_N \cdot p}{\beta + \sum_j g_j(S)}. \tag{3}$$

Under this model, gain variability depends on the same normalization factor as the mean firing rate, and a single new parameter, the noise in the normalization signal $\sigma_N$. While this noise does not depend on the stimulus, the normalization computation causes gain variability to be stimulus dependent.

Qualitatively, this model recapitulates the trends in our data. Increasing stimulus contrast increases the normalization signal and therefore decreases gain variability (Fig. 5b). Increasing stimulus spread has the opposite effect: given a normalization pool composed of narrowly tuned neurons, the normalization signal decreases with spread, thereby increasing gain variability (Fig. 5b).

To test whether this stochastic normalization model quantitatively captures the effects of stimulus uncertainty, we fit the model to half of the data and evaluated its predictions on the other half. Specifically, we fit the only free parameter $\sigma_N$ to the average gain variability measured for the high-contrast stimuli (all other parameters were separately fit to neurons' mean responses, see Online Methods). This single parameter allowed the model to account for the dependency of gain variability on stimulus spread

(Fig. 5c, full line; $P = 0.17$, two-sided absolute goodness-of-fit test). Keeping this parameter constant, we predicted gain variability for the low-contrast stimulus conditions. The model correctly predicted the magnitude of the increase in gain variability (Fig. 5c, dashed line; $P = 0.57$). The uncertainty receptive field could therefore be the functional consequence of a stochastic normalization computation.

**Gain variability exhibits slow dynamics, not fast.** Does gain variability arise from a modulatory process with fast or slow temporal dynamics? If the uncertainty receptive field is the consequence of a stochastic normalization signal, then gain dynamics will follow the dynamics of this signal. The normalization signal arises from a spatial and temporal summation of nearby neural activity[23]. The spatial summation will cause gain variance to track the stimulus energy. However if the stimulus changes slowly (or is constant, as in our experiments) the temporal summation will impart slow dynamics on individual neurons. This would in turn imply that information about stimulus uncertainty can only be transmitted by the joint activity of a sufficiently large population of neurons, not by individual neurons[9]. Crucially, fast and slow modulatory processes have different statistical signatures. If the dynamics are fast, the measured variance-to-mean relation will depend on the duration of the counting window. The larger the counting window, the more within-trial gain variability will be

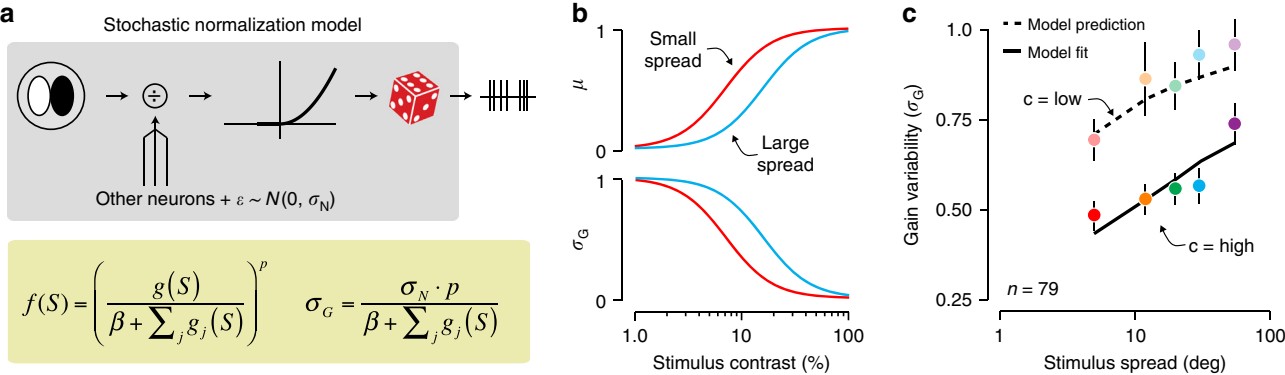

**Fig. 5 A stochastic normalization model accounts for the effects of stimulus uncertainty on gain variability. a** Top: Model diagram. The response of a spatial filter is divided by the summed activity of neighboring units, an additive constant, and a noise source. The normalized signal is passed through a nonlinearity to obtain a firing rate which serves as input for a Poisson process. Bottom: equations for the mean response and gain variability under this model. **b** Mean response (top) and gain variability (bottom) as a function of stimulus contrast for a narrowband (red) and a broadband (blue) stimulus. **c** Measured gain variability compared with the prediction of the stochastic normalization model. The model was fit to the average gain variability across all neurons measured for high contrast stimuli (opaque points), but not to the low-contrast conditions (transparent points). Error bars indicate ± 1 s.e.m.

averaged out, reducing the strength of measured gain fluctuations. In contrast, for a modulatory process with slow dynamics, there is no within-trial gain variability, causing the measured gain fluctuations to be independent of the duration of the counting window[27].

To address this question, we assume that stimulus-independent gain $G$ is constant within temporal intervals of duration $\Delta T$, but varies independently across such intervals. If this duration is longer than all measurement intervals $\Delta t$ (hereafter "slow" dynamics), we recover the variance-to-mean relationship described previously, which is independent of the counting window:

$$\mathrm{Var}[K|S, \Delta t] = \lambda + \sigma_G^2 \lambda^2, \qquad (4)$$

where $\lambda = f(S)\Delta t$ is the mean spike count. In contrast, when $\Delta T$ is smaller than the shortest counting window (hereafter "fast" dynamics), the quadratic term is dampened by the counting window $\Delta t$:

$$\mathrm{Var}[K|S, \Delta t] = \lambda + \sigma_G^2 \lambda^2 \frac{\Delta T}{\Delta t}. \qquad (5)$$

To determine whether gain fluctuations exhibit fast or slow temporal dynamics, we fit these two different versions of the

modulated Poisson model to the same set of neuronal responses. We computed spike counts using differently sized counting windows (Fig. 6a), and then fit the resulting family of variance-to-mean relations imposing either fast or slow dynamics (Fig. 6b). We measured the goodness-of-fit of each model by computing its log likelihood, and then compared both models. A recovery analysis revealed that this method distinguishes fast from slow dynamics with an accuracy of 90.15% (see Online Methods). Each unique stimulus family constitutes one point of comparison for each neuron, yielding a total of 780 data points (78 neurons × 10 stimulus families). Variance-to-mean relations were typically best described as being independent of the counting window. This is evident from the responses of an example neuron. For example, notice how the fast gain dynamics model misses all the data measured with the largest counting window (Fig. 6b, right panel, blue color). Fitting the model exclusively to those data caused it to miss those measured with smaller counting windows (Supplementary Fig. 2). The distribution of log-likelihood differences across the population supports the same conclusion (Fig. 6c; slow dynamics preferred for 85.5% of conditions, median LL difference = −23.4, median LL difference for null model = 2.27, $P < 0.001$, Wilcoxon signed rank test, see

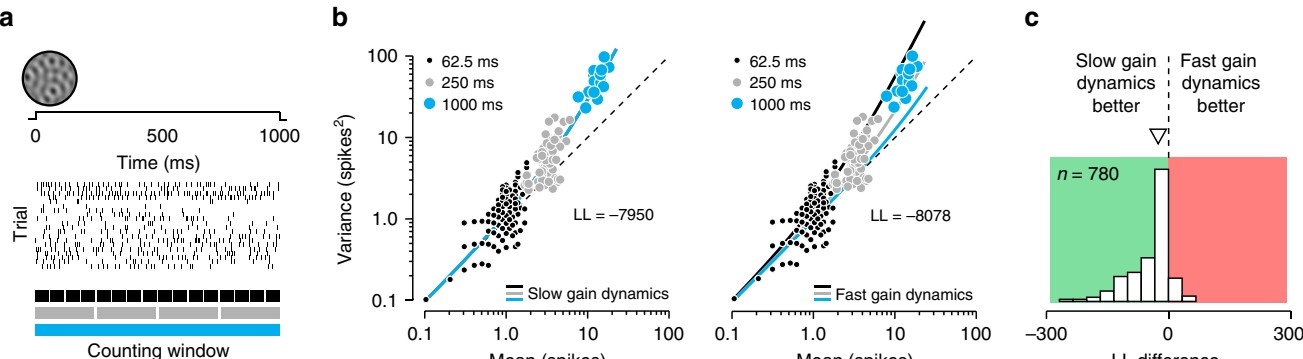

**Fig. 6 Comparison of models with slow and fast gain dynamics. a** For every unique stimulus condition, we computed responses using five different spike-count windows, spanning 62.5, 125, 250, 500, and 1000 ms respectively. **b** Variance-to-mean relation for one stimulus family for an example V1 neuron. Responses are shown for three different counting windows: 62.5 ms (black points), 250 ms (gray points), and 1000 ms (blue points). We fit two models to these data: one with slow gain dynamics (left panel), and one with fast gain dynamics (right panel). Slow gain dynamics predict the same variance-to-mean relationship for different counting window sizes. Fast gain dynamics predict that the variance-to-mean relationship becomes more linear with longer counting windows. We measured goodness-of-fit by computing the log likelihood of the data under each model, yielding a value of −7950 for the slow-dynamics model, and of −8078 for the fast-dynamics model. **c** Distribution of the difference in log likelihood under both models for a population of V1 and V2 neurons.

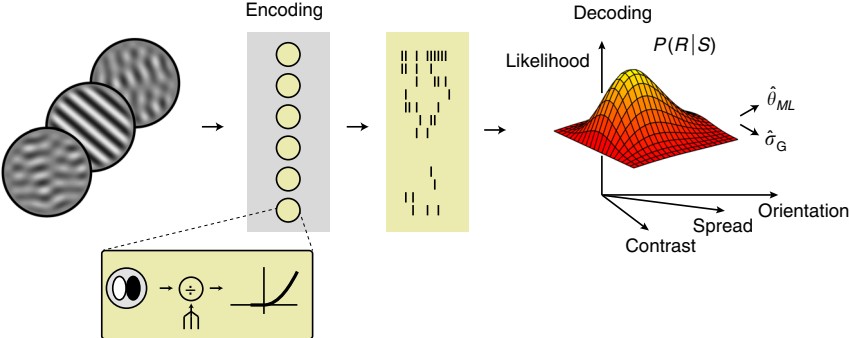

**Fig. 7 Decoding population activity.** We simulated the activity of V1 populations whose mean firing rate and gain variability was governed by the stochastic normalization model. Each unit's stimulus preference and dynamic range was chosen randomly. We used an iterative procedure to decode the most likely stimulus features (orientation, contrast, and spread) from the population activity. The uncertainty receptive field connects these feature estimates to an associated gain variability estimate (Fig. 5b).

Online Methods). In sum, gain variability is much more likely to arise from a slow modulatory process. In such a process, individual neurons communicate a single gain value per trial. Measuring gain variability requires multiple gain values. As a consequence, gain variability cannot be decoded on a trial-by-trial basis from the activity of a single neuron.

**Decoding image features and uncertainty from neural activity.** Organisms have to interpret the environment almost immediately. Sensory circuits must therefore report stimulus features and their associated uncertainty on a moment-to-moment basis. Given that neuronal gain fluctuates slowly, does our proposed coding scheme enable both to be decoded quickly from sensory population activity? We investigated this using model simulations based on our experimental findings. Specifically, we simulated the activity of a population of V1 neurons whose mean firing rate and gain variability resulted from the stochastic divisive normalization model (see Online Methods, Fig. 7). As in cortex, model neurons varied in their orientation preference and dynamic range. For simplicity, we assumed that the magnitude of normalization noise did not differ across neurons. Consequently, the uncertainty receptive field of all neurons had the same tuning, matching our

empirical estimate (Fig. 5c). The model population thus instantiates an idealized version of the neurons we recorded from.

Consider the population response to a briefly presented stimulus (Fig. 7). Stimulus orientation $\theta$ is encoded in the neurons' average response magnitudes $\{\lambda_i\}$, and stimulus uncertainty is represented by cross-neuron variability in response gain $\sigma_G$. We derived the likelihood function for a population of independent, modulated Poisson neurons and used it to determine the maximum-likelihood stimulus estimate (Fig. 7, see Online Methods). This estimate contains the most likely stimulus orientation and, through the uncertainty receptive field, the associated level of gain variability. These estimates $\hat{\theta}_{ML}$ and $\hat{\sigma}_G$ provide a useful indication of how much information regarding stimulus orientation and uncertainty is contained in the population response.

We varied stimulus orientation and uncertainty by manipulating contrast and spread across trials and asked how well each could be decoded from the population response on a trial-by-trial basis. For a population of 250 neurons, stimulus orientation could be decoded near perfectly when stimulus contrast was high (Fig. 8a, red symbols), but less so when contrast was low or the spread was high (Fig. 8a, non-red symbols). This difference in performance was tracked by the simultaneously decoded gain variability. Specifically, when gain variability estimates were low,

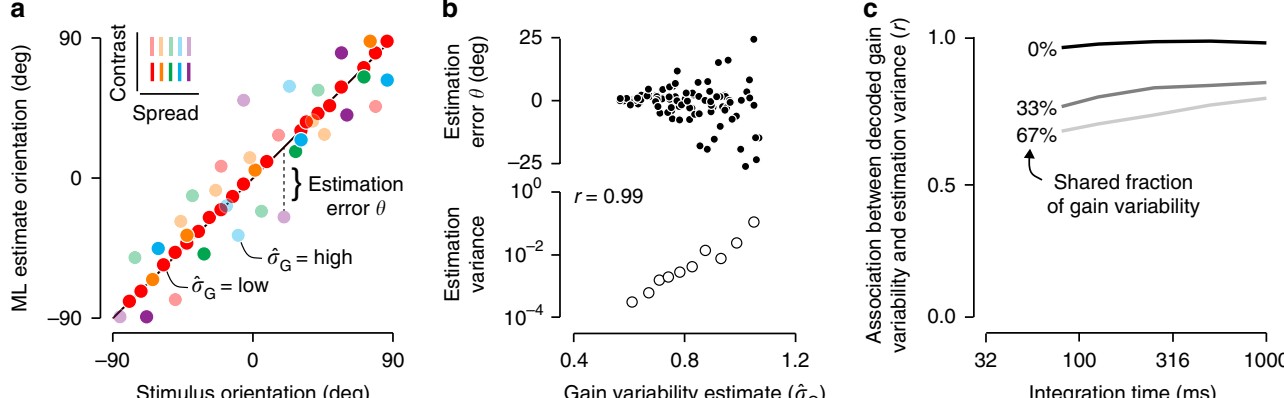

**Fig. 8 Quantitative performance of uncertainty decoding. a** Estimated orientation plotted against stimulus orientation for an example population ($n = 250$ heterogeneous units, integration time = 1000 ms, shared gain fluctuations = 0%). Stimuli varied in orientation, contrast, and spread. **b** Top: orientation estimation error plotted against gain variability estimates. Each symbol represents a single trial. Only a fraction of the trials is shown for clarity. Bottom: the circular variance of the orientation estimation error is plotted against gain variability estimates. Each symbol summarizes 100 trials; trials were grouped based on their gain variability estimates (percentile 0–10, 11–20, etc.). We measure the strength of the association with Spearman's rank correlation coefficient ($r$). **c** Effects of integration time and shared gain fluctuations on uncertainty decoding.

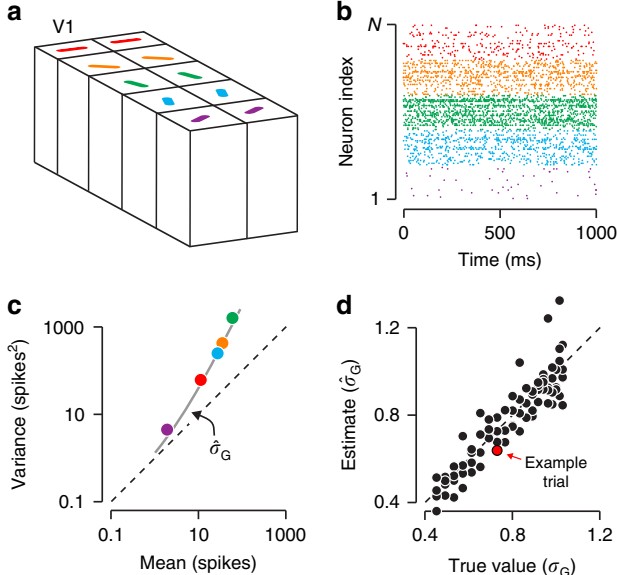

**Fig. 9 Estimating gain variability in a neurally plausible way. a** We simulated the activity of a population of 250 neurons (5 sub-populations of 50 neurons each, loosely based on the concept of cortical columns). All neurons within a sub-population had the same stimulus selectivity, and all neurons had the same uncertainty receptive field, matching our empirical estimate. **b** Simulated population response for a single trial. Color indicates sub-population. **c** Variance-to-mean relation for a single trial. Points summarize responses grouped by sub-population, the line shows the predicted relation under the inferred level of gain variability, using the heuristic estimator. **d** Estimated gain variability plotted against ground truth. Stimuli varied in strength, and hence elicited different levels of gain variability.

the error in orientation decoding tended to be small (Fig. 8b). But when gain variability estimates were high, the error in orientation decoding could be substantial (Fig. 8b; $r = 0.99$). Gain variability estimates thus provide an instantaneously available assay of the reliability of the V1 orientation report.

In the example we considered, the decoder had access to population activity realized over a one-second stimulus epoch. Moreover, all gain variability was statistically independent across neurons, in keeping with our decoder's assumption. Decoding conditions will often be less favorable: fixations typically last only a few hundred milliseconds[28], and gain fluctuations can be partly shared across neurons[15,29,30]. We wondered whether decoded gain variability would still be strongly associated with stimulus uncertainty when read-out time was limited and gain fluctuations were correlated. Fig. 8c illustrates the evolution of this association with read-out time, for different levels of gain correlation. Even under the most challenging conditions—read-out time less than 100 ms and two thirds of gain variance shared across neurons—the association remained substantial (Fig. 8c). We conclude that our coding scheme enables robust decoding of stimulus features and their uncertainty from sensory population activity under physiologically realistic conditions.

How might neural circuits decode gain variability? The maximum-likelihood estimator cannot be computed in closed form and its biological plausibility can therefore be questioned. However there might exist heuristic estimators that only rely on simple, neurally plausible computations. We conceived of one such option. Primate visual cortex exhibits a columnar organization (Fig. 9a). Neurons within the same column share the same

stimulus selectivity and thus constitute a functional sub-population (Fig. 9b). Super-Poisson interneuronal variance within each sub-population can therefore be directly attributed to gain variability (Fig. 9c). This enables estimating $\hat{\sigma}_G$ through a simple heuristic that only relies on common neural computations such as sums, squares, and division (Fig. 9c, see Online Methods). For our idealized population, this heuristic estimator of $\hat{\sigma}_G$ closely tracks the true value (Fig. 9d).

## Discussion

We have proposed a new model of canonical computation in sensory cortex, which incorporates the hypothesis that neurons report features of the environment and the reliability of this message through two different communication channels: the mean spike count and its variance[9]. For example, a change in stimulus orientation might alter the mean firing rate of a V1 neuron, but it will not change its gain variability. A change in orientation noise will alter the neuron's gain variability, but need not change its mean response. We propose that cortical neurons behave as if two different receptive fields underlie these response statistics. We have shown that this behavior naturally arises from known gain-control mechanisms, and does not require an explicit probabilistic inference computation to estimate stimulus uncertainty. We find that gain dynamics are slow relative to behavioral time-scales, hence gain variability cannot be communicated quickly by individual neurons. Nevertheless, we have shown through model simulations that this coding scheme enables sensory populations to rapidly report stimulus features and their uncertainty to downstream circuits, even when gain variability is highly correlated across neurons.

Our framework extends, refines, and potentially bridges two alternative theories for the representation of uncertainty in cortex: probabilistic population codes (PPC), and the sampling hypothesis. The various instantiations of these theories differ in three respects: their use of response variance to represent uncertainty, whether information is represented across time or across neurons, and whether inference is performed in a feedforward manner or through iterative, recurrent computation. In highlighting the importance of gain variability in encoding stimulus uncertainty, our results show that purely mean-based codes[7] cannot provide a full account of the neural representation of uncertainty, and are aligned with the sampling hypothesis[9] in this respect, although this behavior can also arise in non-linear population codes[31–33]. There is some evidence that sensory systems exploit this extra bandwidth. For example, when an observer pays attention to a visual stimulus, perceptual uncertainty can be greatly reduced[34]. In early visual cortex, this behavioral effect is associated with a mild increase in mean response[35], and a comparatively strong reduction in response variability[14]. Moreover, visual attention appears to achieve these effects by employing sensory normalization mechanisms[36,37] and specifically reduces neural gain variability[30,38].

On the other hand, in showing that gain dynamics are slow, our results dispute the notion of temporal representations of uncertainty[9] and are aligned with population-based representations[7], as well as spatial variants of the sampling hypothesis[39]. Our view also differentiates itself from most sampling-based models, which require iterative, recurrent computation to perform accurate inference[40–42]. In contrast, our model can express uncertainty through purely feedforward computations, aligning it with population-based codes and canonical models of neural computation[23,43,44]. Note that our model seeks to describe functional transformations, not the neural mechanisms that implement them—these may rely on recurrent interactions[45]. This conceptual simplicity offers practical benefits, as it allowed

us to straightforwardly fit the uncertainty receptive field to V1 spiking data (Fig. 5c), and to jointly decode stimulus features and their uncertainty from population activity (Fig. 8c). Nonetheless, our feedforward model could be augmented with a recurrent mechanism—in particular to account for behavioral and contextual effects on neural variability[46–49]—an approach that has been shown to combine the advantages of both in machine inference[50].

To test our model, we relied on stimulus manipulations that impair perceptual orientation judgments, and we verified that they reduced the coding capacity of orientation-selective neurons (Fig. 2c, Fig. 3a). Ideally, both sets of measurements would be obtained simultaneously, as this could establish a direct rather than indirect link between neural and behavioral levels. If our model is correct, gain variability should be predictive of errors in perceptual orientation estimates that arise from externally induced stimulus uncertainty. An even stronger test of our framework would be to investigate whether this relationship also holds across repetitions of identical stimuli, where differences in estimation error are solely due to internal noise fluctuations.

Even in the absence of such data, our approach can directly be extended to other stimulus features, visual areas, and sensory systems to investigate the generality of the uncertainty receptive field. As a first step, we have shown that V2 cells, whose mean firing rate is selective for textural properties[21], modulate their gain variability according to uncertainty in visual texture. Crucially, V1 cells, which lack this selectivity, also fail to report this uncertainty. This suggests that, along a sensory processing cascade, selectivity for novel stimulus features and an assessment of their reliability jointly emerge. Why is this so? The sensory neurons that are the first in the hierarchy to represent a particular feature are uniquely positioned to judge the quality of the evidence for that feature. Downstream areas can inherit the feature report, but neural stochasticity entails that uncertainty about this feature can only grow along the hierarchy. Consistent with this, visual areas downstream of V1 exhibit orientation selectivity, but this selectivity is accompanied by systematically increasing levels of gain variability[15].

Our model focuses on gain variability, a specific component of neural response variability. In our framework, alternative measures of response dispersion such as Fano factor do not reflect stimulus uncertainty because they depend on the strength of the stimulus drive and the duration of the count window (Fig. 3d, Fig. 4f, Supplementary Fig. 1). Nevertheless, changes in response gain are a statistical description of neural activity, and are not observed directly. At a mechanistic level they may arise either from fluctuations in neuromodulation or in membrane potential[51]. A new set of measurements, including intracellular physiology, therefore seem necessary to resolve the mechanistic origin of gain variability.

Our results offer a novel view of the structural organization of sensory cortex. Its columnar organization has been known for many decades[16,52,53], yet the computational benefit of this structure has remained elusive[54]. In our coding scheme, estimating interneuronal gain variability is facilitated by the presence of sub-populations of sensory neurons that share the same stimulus selectivity (Fig. 9). In particular, this allows a decoder to infer stimulus uncertainty without detailed knowledge of the sensory neurons' classical receptive field. Whether downstream circuits actually employ this read-out scheme can only be ascertained from an awake, behaving paradigm that requires taking stimulus uncertainty into account. A recent study of this kind found that orientation uncertainty represented by V1 populations (estimated using a flexible, model-agnostic approach) does indeed inform animals' choice behavior[33]. We believe that this paradigm can be leveraged to test our and other theories, and will ultimately

uncover which aspect of neural activity informs perceptual uncertainty estimates.

Finally, our results reveal a strong connection between biological and machine inference under uncertainty. Recent years have witnessed the development of a new class of highly scalable artificial inference methods[55,56]. Like our coding scheme, these methods forfeit exact inference which often requires costly iterative procedures[57] in favor of simple, parametric approximations that can be computed in a feedforward manner. The resulting efficiency and scalability have enabled progress in highly complex problems such as scene understanding[58], autonomous navigation[59–61], and robotic manipulation[62]. Biological systems face similarly complex tasks and environments, and may also have opted for inference methods that are simple and powerful.

## Methods

**Physiology**. The data analyzed here were previously published, and the full methods are provided there (see ref. [17] for the orientation experiment, and ref. [22] for the texture experiment). In brief, all recordings were made from anesthetized, paralyzed, adult macaque monkeys. Surgical preparation methods are reported in detail in (ref. [63]). Anesthesia was maintained with infusion of sufentanil citrate (6–30 g kg$^{-1}$ h$^{-1}$) and paralysis with infusion of vecuronium bromide (Norcuron; 0.1 mg kg$^{-1}$ h$^{-1}$) in isotonic dextrose-Normosol solution. All experiments were conducted in compliance with the NIH's *Guide for the Care and Use of Laboratory Animals*, and with approval of the New York University Animal Welfare Committee. Extracellular recordings from individual neurons were made with quartz-platinum-tungsten microelectrodes (Thomas Recording), advanced mechanically into the brain through a craniotomy and small durotomy. V1 was distinguished from V2 on the basis of depth from the cortical surface and changes in the receptive field location of the recorded units.

**Visual stimulation**. In the orientation experiment, stimuli consisted of Gaussian orientation mixtures, created by summing nine sinusoidal gratings whose orientations were spaced at 20° intervals and whose orientation-dependent contrasts followed a circular Gaussian profile centered on a particular orientation (spread 0–55°). The drift rate of each stimulus component was selected at random from a Gaussian distribution centered on the preferred rate, with a standard deviation equal to 1/5 this value, resulting in an incoherently drifting mixture. In total, ten stimulus families (five spread levels × two contrast levels) were presented at 16 different orientations.

In the texture experiment, stimuli were generated using the texture analysis-synthesis procedure introduced by[64]. Fifteen different grayscale photographs of visual texture served as prototypes. From each of these source images, two sets of 15 samples were synthesized (one set of "naturalistic textures", and one set of "unstructured noise stimuli"). The naturalistic textures preserved the spectrum of the original image, as well as correlations across the output of filters tuned to different positions, scales, and orientations; the noise stimuli preserved only the spectrum[22].

In both experiments, stimuli were presented in random order for either 1000 ms (orientation experiment) or 100 ms (texture experiment), and typically repeated 10 times (orientation experiment) or 20 times (texture experiment).

**Data analysis**. For all analyses of the orientation experiment but one, we counted spikes within a 1000 ms window following response onset. One analysis sought to compare spiking models with slow vs fast gain dynamics (Fig. 6). Here, we used five different counting windows (62.5, 125, 250, 500, and 1000 ms). For the analysis of the texture experiment, we computed spike counts using a 100 ms window aligned to the response onset.

**Quantifying neural stimulus uncertainty**. Using standard tools from information theory[20], we quantified neural stimulus uncertainty in the orientation domain as the inverse of a neuron's Fisher Information for a given stimulus family. If neural responses arise from a Poisson process, this statistic can be simply written as a function of the measured tuning curve $h(\theta)$:

$$\frac{1}{I_\theta} = \mathrm{E}_\theta \left[ \frac{h'^2(\theta)}{h(\theta)} \right]^{-1}, \tag{6}$$

where $h'(\theta)$ is the derivative of the tuning curve (ref. [65]). This statistic has the benefit that its value only depends on the measured mean responses, and is independent of the level of gain fluctuations. Associations between gain variability and stimulus uncertainty (Fig. 2f, Fig. 3b, c) can thus not arise for trivial reasons. This is not true of alternative estimators of uncertainty which rely on empirical measurements of response variance rather than a Poisson assumption.

**Measuring gain variability**. We measured gain variability using the method introduced by ref. [15]. Specifically, we described responses of individual neurons with a model in which spikes are generated by a Poisson process whose rate is the product of a stimulus-dependent drive and a stimulus-independent gain. We assumed that gain is constant within a trial and distributed across trials according to a gamma distribution with mean 1 and variance $\sigma_G^2$. We estimated this parameter by maximizing the likelihood of the full set of observed spike counts for a given stimulus family under a negative binomial distribution[15] (Fig. 2d, Fig. 4c).

We computed the selectivity of gain variability for induced stimulus uncertainty (Fig. 3d, Fig. 4d) by taking the common logarithm of the ratio of two $\sigma_G$ estimates: one measured in the presence of the uncertainty-inducing manipulation (numerator), and one measured in its absence (denominator). For the texture experiment, we performed a significance test on this statistic (Fig. 4c, inset). For each neuron, we obtained a null distribution by first estimating gain variability from the combination of all stimulus conditions. Next, we used this value and the empirically observed mean responses to simulate 100 synthetic datasets. For each synthetic dataset, we then separately estimated gain variability for responses to texture and noise stimuli. We used these values to compute the distribution of the selectivity-index to be expected if there were no underlying difference in gain variability between texture and noise stimuli (estimated from $100^2$ samples per neuron). Because gain variability is a positive-valued statistic, estimation error can introduce a bias that depends on the magnitude of the mean response. Consequently, the null distribution need not be centered at zero. We deem the empirically obtained selectivity value significant if it falls outside of the central 95 percent interval of this distribution.

**Measuring Fano factor**. We examined the relationship between stimulus uncertainty and Fano factor, a popular measure of response dispersion, defined as the ratio of the spike-count variance to the mean. This statistic does not capture a stable property of a neuron for a given level of stimulus uncertainty, as it depends on stimulus drive and count window (Supplementary Fig. 1). To obtain a single value of Fano factor for each stimulus family in the orientation experiment, we first computed an estimate for each stimulus condition and then averaged these estimates across all stimulus orientations within a given family (Fig. 3d). Likewise, in the texture experiment, we averaged the condition-specific estimates across all texture and noise stimuli, respectively (Fig. 4f). To obtain a single value of Fano factor across conditions, some previous studies used a different computation which takes into account the statistical uncertainty of the response variance estimates[10].

**Fitting the stochastic normalization model**. The canonical divisive normalization model describes the deterministic firing rate $f_i(S)$ of a neuron $i$ in response to a stimulus $S$ as some function of the stimulus drive $g_i(S)$ divided by the sum of stimulus-dependent drive to neighboring neurons $\sum_j g_j(S)$ and a stimulus-independent constant $\beta$, with transduction exponent $p$:

$$f_i(S) = \left( \frac{g_i(S)}{\beta + \sum_j g_j(S)} \right)^p. \tag{7}$$

Because neighboring neurons are stochastic, we modeled the aggregate stochasticity of the normalization pool with stimulus-independent additive Gaussian noise $\epsilon \sim \mathcal{N}(0, \sigma_N^2)$ and define the resulting stochastic firing rate:

$$\mu_i = \left( \frac{g_i(S)}{\beta + \sum_j g_j(S) + \epsilon} \right)^p. \tag{8}$$

If the magnitude of the noise $\epsilon$ is sufficiently small, we can use a Taylor expansion to obtain the mean and standard deviation of the firing rate $\mu_i$ across samples of normalization noise:

$$\mathrm{E}[\mu_i] = f_i(S), \tag{9}$$

$$\mathrm{Std}[\mu_i] = \frac{\sigma_N \cdot p}{\beta + \sum_j g_j(S)} f_i(S). \tag{10}$$

Equating these expressions to those obtained from the modulated Poisson model (recall $\mathrm{E}[\mu_i] = f(S)$, $\mathrm{Std}[\mu_i] = f(S)\sigma_G$) results in a new expression for gain variability:

$$\sigma_G = \frac{\mathrm{Std}[\mu_i]}{\mathrm{E}[\mu_i]} = \frac{\sigma_N \cdot p}{\beta + \sum_j g_j(S)}. \tag{11}$$

Although the noise term $\sigma_N$ is stimulus-independent, divisive normalization causes gain variability to depend on the stimulus through the denominator of this expression.

We investigated the adequacy of this equation by fitting the stochastic normalization model to the population-averaged gain variability. We opted to constrain the model as much as possible. Rather than fitting the transduction exponent $p$ and the stimulus-independent normalization constant $\beta$ to these data, we used the population-averaged estimates of both parameters obtained by fitting the neurons' mean responses with the divisive normalization model from ref. [17] ($p = 2.001, \beta = 0.64$). We approximate the exponent with $p = 2$ to align our model with canonical formulations of divisive normalization[24]. The stimulus-dependent

normalization $\sum_j g_j(S)$ was computed by simulating responses of a fixed pool of neurons with a diverse set of tuning properties, as explained in detail in ref. [17].

The final free parameter $\sigma_N$ was estimated by minimizing the mean squared error between predicted and observed $\sigma_G$ (Fig. 5c, full line).

**Analysis of gain dynamics**. We sought to determine whether neural gain fluctuations are better described as having fast or slow dynamics. For a slow modulatory process, the variance-to-mean relationship is independent of the counting window; for a fast process, this relation changes in a predictable manner with window size (see equations in Results). To leverage this insight, we counted the same set of spikes with windows of different duration, and fit both a fast- and a slow-dynamics model to the resulting dataset. The largest counting window (1000 ms) contributes one observation per trial; the smallest window (62.5 ms) contributes sixteen observations per trial. To determine the log likelihood of the models for an entire dataset, we treat all observations as being statistically independent. This is not strictly correct, as each spike is counted multiple times (exactly once per window size). To assess the effectiveness of our model comparison procedure, we performed a recovery analysis. For each measured variance-to-mean relation (one per neuron per stimulus family), we synthesized 1000 datasets imposing slow gain dynamics (one random gain sample per second), and 1000 datasets imposing fast gain dynamics (one random gain sample every 62.5 ms). The generating parameters were the empirically observed mean counts as measured with a 62.5 ms window, and the gain variability estimate obtained with a 1000 ms window. We then fit the slow- and fast-dynamics model to each synthetic dataset, and compared their goodness-of-fit in exactly the same manner as we did for the real data. When the ground truth was slow dynamics, the slow-dynamics model was preferred in 99.5% of cases; when the ground truth was fast dynamics, the fast-dynamics model was preferred in 80.8% of cases. We deem our method to be fairly sensitive, although slightly biased in favor of slow dynamics. If slow and fast dynamics were equally probable in the population, our method would identify the slow-dynamics model as the winner in 59.4% of cases. In contrast, when applied to real data, slow dynamics were favored in 89.10% of cases. To assess the significance of this difference, we compared the empirically obtained log likelihood difference with the expected log likelihood difference under a null model, created by combining all 2000 synthetic datasets (and thus making slow and fast dynamics equally probable), and found slow dynamics to be preferred in 85.5% of cases.

As an additional control, we also performed an analysis in which we only fit the modulated Poisson model to the largest counting window conditions, and then generated predictions for all other window sizes assuming either fast or slow dynamics. For 88.85% of cases, the slow-dynamics model generated better predictions than the fast-dynamics model (Supplementary Fig. 2).

**Decoding stimulus features and uncertainty**. Stimulus orientation, spread, and contrast were jointly decoded on a trial-by-trial basis from simulated population activity. We defined stimuli $S$ in the orientation domain as mixtures consisting of up to nine components that were spaced at 20° intervals and whose orientation-dependent contrasts followed a circular Gaussian profile centered on a particular orientation $\theta_S$. Stimulus spread $\sigma_S$ was varied between 1–55° and stimulus contrast $c_S$ (i.e., the amplitude of the Gaussian) between 5% and 50%. Stimuli were processed by a population of neurons whose orientation selectivity $W_i$ was determined by a raised cosine function:

$$W_i(\theta) = \cos^3(\theta - \theta_i) \exp\left( \frac{9}{2} \cos^2(\theta - \theta_i) \right), \tag{12}$$

where $\theta_i$ is the preferred orientation. This profile matches the selectivity of a spatial Gaussian derivative filter with an aspect ratio of two and derivative order of three[17]. Stimulus drive $g_i(S)$ was computed as the dot-product of the stimulus and filter profiles, followed by an affine rescaling:

$$g_i(S) = \eta + \upsilon \sum_\theta W_i(\theta) \cdot S(\theta), \tag{13}$$

where $\eta$ captures the spontaneous discharge and $\upsilon$ the dynamic range (i.e., the difference between the spontaneous discharge and the response elicited by the preferred stimulus). We then applied the equations of the stochastic normalization model to obtain a firing rate $f_i(S)$ and gain variability $\sigma_G$ for each neuron (Eqs. 1 and 2). Populations consisted of 250 neurons, and each neuron's orientation preference and dynamic range were chosen randomly from a uniform and Gaussian distribution, respectively. The spontaneous discharge $\eta$ equalled 2 ips on average (s.d.: 0.2 ips), and the dynamic range $\upsilon$ equalled 50 ips on average (s.d.: 7 ips). All neurons had the same uncertainty receptive field whose shape was determined by parameters fit to neural data ($\sigma_N = 0.35, p = 2, \beta = 0.64$), resulting in a single value $\sigma_G$ for the gain variability of the entire population.

Assuming these neurons fire independently from one another, we modeled a pattern of spike counts $\{K_i\}$ from a window of length $\Delta t$ using a negative binomial

distribution[15]:

$$\log p(\{K_i\}|S) = \log \prod_{i=1}^{n} p(K_i|S)$$

$$= \sum_{i=1}^{n} \log \Gamma(K_i + 1/\sigma_G^2) - \log \Gamma(K_i + 1) \qquad (14)$$
$$- \log \Gamma(1/\sigma_G^2) + K_i \log (\sigma_G^2 \lambda_i)$$
$$- (K_i + 1/\sigma_G^2) \log (1 + \sigma_G^2 \lambda_i),$$

where the rate $\lambda_i = f_i(S)\Delta t$ and gain variability $\sigma_G$ are given by the stochastic normalization model (Eqs. 2 and 3). This allowed us to compute the most likely stimulus given a collection of spike counts $\{K_i\}$:

$$\hat{S} = \arg\max_{S} \log p(\{K_i\}|S). \qquad (15)$$

In particular, given that in our case the stimulus was fully defined by its peak orientation $\theta_S$, its contrast $c_S$ and spread $\sigma_S$, we simultaneously decoded these variables via maximum-likelihood estimation:

$$\hat{\theta}_S, \hat{c}_S, \hat{\sigma}_S = \arg\max_{\theta_S, c_S, \sigma_S} \log p(\{K_i\}|S(\theta_S, c_S, \sigma_S)). \qquad (16)$$

We found this maximum-likelihood estimate via gradient ascent using fmincon in MATLAB (using a multi-start procedure with random initialization) while constraining the stimulus estimates to be within the following ranges (contrast: [0, 1], orientation: [0°, 180°], spread: [0°, 70°]). Having done so, we compute an estimate of the gain variability by evaluating the uncertainty receptive field on the estimated stimulus parameters. This is the decoded gain variability reported in Fig. 8.

To assess the quality of uncertainty and orientation decoding, we measured the orientation decoding error on a trial-by-trial basis (Fig. 8a). Each simulation included 100 unique contrast-dispersion stimuli at ten orientations, yielding a total of 1000 trials. We sorted and binned the trials according to the estimated gain variability $\hat{\sigma}_G$. Within each bin, we computed the variance of the orientation estimation error across trials, and compared it to the average gain variability estimate of that bin (Fig. 8b). The reported association between these two quantities (Fig. 8c) is their Spearman correlation, averaged across 100 repeats of the simulation.

To assess the effect of interneuronal gain correlations, we varied the amount of gain correlation while keeping the total amount of gain variability constant. Specifically, we created two gain variables $G_s$ and $G_p$ that were shared and private respectively, both of which had unit mean and a variance equal to $\sigma_G^2$. Each neuron was modulated by its own gain $G = \gamma G_s + (1 - \gamma) G_p$ where $\gamma \in [0, 1]$. When $\gamma = 0$, all gain variability is statistically independent across the population, when $\gamma > 0$, interneuronal gain fluctuations are positively correlated. We chose $\gamma \in \{0, 0.33, 0.67\}$ to span a physiologically plausible range[15,66]. Finally, we wished to estimate gain variability in an efficient, neurally plausible manner. For this we make the additional assumption that our population of neurons is divided into $n = 5$ sub-populations (or cortical columns) of $m = 50$ neurons who share identical stimulus tuning $\lambda_i$. In this case, firing rates can be estimated by averaging the spiking counts $K_i^j$ within a sub-population:

$$\lambda_i \approx \hat{\lambda}_i = \frac{1}{m} \sum_{j=1}^{m} K_i^j \qquad (17)$$

with the approximation becoming exact in the limit of a large sub-population size $m$. Similarly, the variance within sub-populations provides an estimate of their true variance, and thus the gain variability:

$$\lambda_i + \sigma_G^2 \lambda_i^2 = \sigma_i^2 \approx \hat{\sigma}_i^2 = \frac{1}{m-1} \sum_{j=1}^{m} (K_i^j - \hat{\lambda}_i)^2. \qquad (18)$$

If we further assume that gain variability is shared across sub-populations, we can pool these estimators into a single estimate of gain variability for the entire population:

$$\sigma_G^2 \approx \hat{\sigma}_G^2 = \frac{\sum_{i=1}^{n} \hat{\sigma}_i^2 - \hat{\lambda}_i}{\sum_{i=1}^{n} \hat{\lambda}_i^2}. \qquad (19)$$

This is the heuristic estimator shown in Fig. 9.

**Reporting summary**. Further information on research design is available in the Nature Research Reporting Summary linked to this article.

## Data availability
The data and analysis code that support the findings of this study are available from the corresponding author upon reasonable request.

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

## Acknowledgements

This work was supported by a Whitehall Foundation grant (OSP No 201900549 to R.L.T.G), an NSF-GRFP (no. 000392968 to Z.M.B.S), and an N.I.H. training grant (T32 EY021462 supported C.M.Z.). We wish to thank Xaq Pitkow for valuable discussions.

## Author contributions

O.J.H. and R.L.T.G. conceived the project and developed the theoretical framework. Z.M.B.S. performed all data analyses and simulations for the orientation experiment. K.M. assisted with data analyses for the orientation experiment. C.M.Z. performed the data analysis for the texture experiment. O.J.H. and R.L.T.G. wrote the manuscript, with contributions from all authors.

## Competing interests

The authors declare no competing interests.
