## [Peer Review File · Nature Communications]

Reviewers' Comments:

Reviewer #1:

Remarks to the Author:

This manuscript suggests to provide a new theory of the representation of uncertainty in the visual cortex. According to this view, "...average neural response strength encodes stimulus features, while cross-neuron variability in response gain encodes the uncertainty of these features...". To support their claims, the authors analyse anesthetized macaque single cell recording from V1 and V2, and make several observations based on their proposed model that handles both mean values and uncertainty of the neural signal.

While the topic of the paper is worthwhile, I have several problems with the logic and the delivery of the ideas presented in the paper. I am going to list these by contrasting the manuscript to the paper of Orban et al. (2016, *Neuron*), first since the Orban et al. paper has the most comprehensive treatment of the issue of variability and uncertainty for neural coding to date, and second because the authors themselves compare their work to that paper.

The original work (Goris et al, 2014, *Nat Neu.*) which serves as a conceptual basis of the present paper introduced the idea of a gain signal G , which is independent of the sensory input, and as such, brings in an extra term of variability. With an appropriate fitting of the parameters of this extra term, the joint variability of this "modulated Poisson" process could capture the supralinear nature of the variance-to-mean relation found in anesthetized recordings. This modulated Poisson model is a generic descriptive model that does not specify – beyond some general terms, such as "attention" – the origin or the fine structure of gain variability other than pegging it to a gamma distribution.

In the present manuscript, the authors move forward and look for the source of the gain signal and its variability. They embrace the concept that mean firing rates and variability of cell responses are separate and independent measures related to environmental features and uncertainty in encoding those features, respectively. However, the authors seem to embrace this idea a bit too strongly. Throughout the manuscript, it feels as they imply that this is an original contribution of the paper. For example, the starting line of the discussion says: "We have introduced a new perspective on the neural code. In our view, sensory neurons report the features of the environment and the reliability of this message through two different communication channels: the mean spike count and its variance. For example, a change in stimulus orientation might alter the mean firing rate of a V1 neuron, but it will not change its gain variability."

In fact, the concept that "average neural response strength encodes stimulus features, while cross-neuron variability in response gain encodes the uncertainty of these features" (c.f. abstract of this manuscript) has been prominently proposed in Orban et al. (2016) together with a particular implementation of this concept through neural sampling, and the proposed model was extensively tested against a large set of data including the example above with the orientation-invariance of response variability. Thus, the contribution of the present manuscript is not introducing the new perspective of separable coding of mean values and uncertainty, but an implementational trick, postulating an "uncertainty receptive field" in order to encoding some measure of uncertainty separately. While Orban et al. (2016) used sampling for this based on the concept of coding the joint posterior, the present paper uses a PPC-like approach with doubling the system with positing a second set of RF structure. It is fine to explore this implementation and link the gain concept to uncertainty this way, but the entire manuscript should do a much better job clarifying what is their own contribution and what are the ideas incorporated from previous research, especially when the given research has been referred to in the manuscript.

Once the truly new ideas of the manuscript are identified, the next question is to see the solidity and contributions of these ideas that advance our understanding beyond what was known before. Here, I have the following questions.

Regarding the concept of the uncertainty RF (URF), it is unclear what this URF really represents. The descriptive model of Goris et al (2014) partitioned variability in a mechanical sense: some variability comes from the sensory input and some other comes from other sources (top-down effects, local connection, etc.). This is fine for approximating measured variability, but tying this model conceptually to uncertainty is problematic. Should we take the authors statement "We propose that, while average response magnitude encodes stimulus features, variability in response gain encodes the uncertainty of these features. (p. 1)" literally? In other words, is it only the part of variability delivered by the gain term that defines uncertainty in the system and the point-process variability does not matter? If so what does this measure represent? It cannot represent the cell's full uncertainty about the interpretation of the stimulus since that would clearly require considering the quality of the input information. If the argument is that momentary gain variability essentially incorporates input variability of the previous moment (as suggested by the authors' wording e.g. p.2), then the separation into gain variability and input variability is unnecessary. In this case, the bona fide measured variability of the cell response that counts, the concept of URF is unnecessary, and the proposition of the manuscript is very much the same as that of Orban et al. (2016): a measurement of variability conveys information about uncertainty. The results and contributions of the manuscript in this case can be summarized as this: taking the restricted conditions of anesthetized animals and orientation selectivity, when neural spiking behaviour is modelled by the Goris et al (2014) model, the variability of the Gain term closely follows the true uncertainty of the cell about the stimulus, and so a hypothesised summary measurement of this variability, i. e. the width of the URF gives similar results to other measurements using a more comprehensive measure of the full variability. Under these conditions, variability can also be approximated with simple divisive normalization.

If this characterization is correct, two important questions are whether the present model has distinguishable behaviour from other models and whether that distinguishing behaviour provides additional benefits in terms of deeper understanding of neural coding. Since, it is unclear how the measure of variability conveyed by the URF compares to other measures of variability in the cell response, it would be very beneficial, if the authors could clarify this and list features that set apart their model from others. If, as it was presented in the discussion, the authors suggest that their "width of the distribution" parameter is a quick-and-dirty, but efficient feed-forward first approximation of the ongoing true uncertainty, it would be very nice to spell out the range of its applicability. Specifically, under what conditions is this mode sufficient? Is it useable in natural situations? Is it necessary to assume this separate mechanism for the feed-forward pass which is augmented by other uncertainty representations (c.f. active vs. passive comparison in Discussion p. 5), or it can be viewed as re-describing the initial segment of the more complete uncertainty estimation process used in active vision with complex natural conditions? If these are separate mechanisms, how do they work together, is there a continuous combination of information, a hand-shaking/take-over between the two mechanisms, if so, when and how? Answering these questions would greatly enhance the impact of the paper.

One benefit would be if the model could be extended beyond the tested data set. From this point of view, Orban et al. (2016) delivered an analysis showing how some aspects of the structure of variability can be tested in anesthetized animals while other cannot. In the same way as classical receptive fields have diminishing descriptive power in awake animals performing a complex task, it is a question how well the proposed URF scheme hold up in such conditions. A test with such a data set would be illuminating.

A prominent claim of the manuscript is that the benefit of the URF formalism is easy decodability. In the Orban et al (2016) paper, an analysis was provided showing that measuring orientation with the nuisance parameter of contrast is special in the sense that this is almost the only setup where linear decoding could operate close to optimal (cf. Suppl Mat). Selecting any other combination of feature/nuisance e.g. feature= orientation and nuisance= phase or aperture, the linear decoding falls short of optimal decoding sometimes as much as by 40%. Can the present model be extended to nonlinear decoding and can uncertainty be computed efficiently in that situation?

Jozsef Fiser

Reviewer #2:

Remarks to the Author:

How do neural circuits represent uncertainty? This is one of the biggest open questions in systems neuroscience, and this paper does a beautiful job at starting to resolve this question in the macaque early visual cortex. In a nutshell, the authors find that it is the variability of neural responses that encodes uncertainty, this variability is well captured by variability in neural gain, and even though it can only be read out on long time scales from single neurons, it can still be read out efficiently on short time scales from a population of neurons. These are important findings on their own right, but also because they offer important support for one particular theory of the neural representation of uncertainty over alternative theories (see more on this below). Overall, the paper is very well written with very informative (and highly aesthetic!) figures, thorough analyses and an impressive combination of theory and experiments. I am confident that it will be a must-read in the field. Nevertheless, I would like the authors to address some important points in the final version.

I.

It is precisely because of the predicted high status of this paper in the field that I think the authors have a special responsibility of interpreting their results carefully in the context of the literature on neural representations of uncertainty, in particular the two competing theories that they single out: PPCs and sampling. [Disclaimer: as the authors will have guessed, this reviewer is a proponent of the latter — see my signature below.] While the Introduction clearly states that the paper offers a concrete example of sampling ("Here we incorporate this hypothesis [of Ref. 9 — a key paper on sampling] in the canonical model of neural coding"), the Discussion tries to give a more "balanced" view, trying to keep equal distance from the two camps. While being balanced is the right thing to aim for, here I think the authors reach a somewhat false balance, based on (and perpetuating) a misunderstanding of the main difference(s) between PPCs and sampling. As Ref. 9 argues, the main difference is that in PPCs (and some other representations) "both the mean of a probability distribution and the associated uncertainty are represented by time-average neural responses", while in sampling "the mean and variability of responses [serve] as independent information channels, respectively encoding the mean and the associated uncertainty of the probability distribution over visual feature". Compare this to how the authors of this paper characterise the main point of their paper (second sentence of the Discussion): "In our view, sensory neurons report the features of the environment and the reliability of this message through two different communication channels: the mean spike count and its variance." I think, based on this, it is pretty clear that the author's results fall squarely within the realm of sampling.

It is true that there is a secondary difference between PPCs and sampling: that PPCs are "instantaneous" (inasmuch as mean responses can be quickly read off), while sampling is usually

thought of as a process evolving through time. However, it is a general (and by now mostly self-evident) rule in neural coding (of about anything) that you can buy time by neurons (or vice versa, if neurons are the more limited commodity) by "averaging" (computing statistics) across neurons rather than time. Indeed, in the specific context of sampling, Ref. 9 explicitly mentions this possibility: "Although, by using over-complete representations, in which many neurons effectively code for the same variable [which is precisely the case the authors explore in this paper in Figs. 7-8!], even one sample of a population activity pattern may represent multiple samples of the relevant variables, such that the effective rate of sampling can be faster than expected from neural time constants", and Savin and Deneve (cited in Ref, 9) provide a worked out example for this (again, in the context of sampling).

There is a third (seeming) difference between PPCs and sampling that the authors emphasise: that PPCs encode parametric distributions, while sampling represents "an arbitrarily flexible posterior distribution". However, strictly formally, samples can also be interpreted in a parametric form, as a mixture of deltas. More importantly, in the context of neural representations, the posteriors that need to be represented are themselves already considered to be a highly constrained parametric family (eg Gaussian in the Gaussian scale mixture model, as in Ref. 9), and once neural circuit mechanisms are proposed, these also obviously place severe constraints on the parametric form of distributions that can be represented (eg Buesing et al, PLoS Comput Biol 2011; Hennequin et al, NIPS 2014; Aitchison et al, PLoS Comput Biol 2016 [both Gaussian]; or Echeveste et al, bioRxiv 2019). (So, technically, in ML terms these circuits perform amortised variational inference while using samples to represent the resulting posteriors.) These parametric constraints also mean that should decoding (of uncertainty or else) be necessary, it can also be greatly simplified (and potentially sped up, see above) by taking them into account appropriately. Therefore, this distinction is largely a red herring, at least without a more nuanced discussion.

So, I have two (plus one) concrete suggestions for how to present this whole issue in the Discussion more fairly. First, I feel the first sentence of the Discussion is somewhat overstated by claiming a "new perspective on the neural code" which is supposed to refer to the idea of the "two different communication channels" described above. The same goes for the sentence later in the same paragraph stating that "this is a new conceptualization of the neural code". As the quote from Ref. 9 above shows, this idea as such is very much the same as that in Ref. 9 (and related papers) and the paper is strong enough without making this claim — so I would present this more in the way the Introduction does, saying that the paper presents (and tests!) a specific, novel instantiation of the sampling framework. Second, when discussing the PPC vs sampling distinction, and the results of this paper in the context of these, make clear that there are these two different "axes" along which they differ, with the second (time vs "space") being secondary to the first (same vs different information channels for communicating mean and uncertainty). I would either not mention the third axis (parametric vs "arbitrary" distributions) or with the important disclaimers mentioned above.

II.

In practical terms, given the arguments above, I have a couple of questions and suggestions for the decoding part.

1. Fig. 7b shows the orientation tuning curves of the simulated population. Please also state explicitly (and preferably show) how, if at all, mean responses depended on contrast.
2. At present, only one decoder, an approximation of the optimal maximum likelihood decoder, is studied. In order to establish how important the "second information channel" of variability (see above) is, it would be useful to construct a decoder that only uses the mean responses and quantify how it fares relative to the current one that makes use of response variability. In addition, along the

lines of Ref. 9, a proper comparison with a (linear) PPC (to see how important contrast- or other “nuisance” parameter-dependent modulations of variability — ignored by PPCs — are) requires training a linear decoder at one contrast (nuisance) level (or with contrast unknown) and again comparing its performance with that of the current decoder.

3. Perhaps it’s worth considering repeating these analyses with the other main modulator of uncertainty considered in the paper: stimulus spread.

III.

I have two more relatively major comments regarding two central findings of the paper.

1. A central finding of the paper is that changes in above-Poisson variability are best characterised as changes in the response gain of neurons. However, these gain changes are inferred from patterns of variability, not observed directly. An alternative hypothesis is that above-Poisson variability is caused by fluctuations in firing rate which in turn are due to fluctuations in membrane potential, or the total input to cells (Hennequin et al, Neuron 2018), and uncertainty could be represented by these fluctuations instead (as in Ref. 9 or Echeveste et al, bioRxiv 2019). These alternatives might be distinguishable from the gain fluctuations-based mechanism proposed here, based on the precise scaling of spike count variance with mean spike count and with the measurement time window (ie the kind of analyses performed in the paper). Some quick back-of-the-envelope calculations based on Hennequin et al (arXiv, 2014) suggest that a standard rectified power law nonlinearity mapping from membrane potentials to instantaneous firing rates, with the appropriate choice of the power law exponent and limit in the time scale of fluctuations (fast vs slow), may be able to result in the right scaling of spike count variability with either factor but perhaps not both simultaneously. In any case, this potential alternative should at the very least be discussed, if not analysed properly.

2. Another key finding is that gain fluctuations are slow(er than ~ 1 s). I think some discussion would be due here as to how much this may be specific to the anaesthetised case, where slow fluctuations are known to dominate which are in turn largely absent in the awake V1 (Ref. 29, but to some degree also Ref. 15 for MT by the senior author).

Minor comments

- p.1, Results, para 1: “The optimal estimation strategy” I think some qualification is needed here as maximum likelihood described here is not universally optimal.
- p.3, para 1 of “The uncertainty receptive field as a consequence of stochastic divisive normalization”: “Finally, divisive normalization can be instantiated in image-computable models” Please expand on what you mean by “image-computable models”.
- p.3, last para of “The uncertainty receptive field as a consequence of stochastic divisive normalization”: “Specifically, we fit the only free parameter” I assume that this means that beta and p have been separately fitted to the means, that’s why at this point sigma_N is the only free parameter left. Please explain this explicitly.
- p.9, para 1: “we synthesized 1,000 data-sets imposing slow gain dynamics, and 1,000 datasets imposing fast gain dynamics” Please specify what time scales were used exactly in the two cases, and what was the functional form of the dynamics (e.g. AR(1)?). Instead of performing model selection between two extreme cases (“slow” and “fast”), could you have fitted quantitatively the time scale of dynamics? Please explain your rationale.
- p.9, eq. for Std[lambdai]: There seems to be a factor of p missing from the RHS.
- p.9, para after eq. for sigma_G: “fitting the neurons’ mean responses with a divisive normalization model ($p = 2$, $\beta = 0.64$)” How come you got precisely an integer value for p? Is this because you only allowed integers? Please explain.
- p.9, para above last eq.: “given a collection of $n \times m$ spike counts” Please use proper multiplication

sign instead of asterisk.

- p.9, last eq.: Shouldn't there be a Δt term multiplying $f_i(S)$ whenever it appears?
- p.10, description of the heuristic estimator: Perhaps it's worth noting that maximising the (marginal) likelihood of σ_G and then maximising the likelihood of S having fixed σ_G at this value does not generally give the same solution as jointly maximising the likelihood wrt. both parameters simultaneously — even if these maxima can be computed exactly.
- p.11, caption of Fig.1: The "Hi Olivier" text seems unintended.
- Fig.2b: Please also show example tuning curve when stimulus spread is changed.
- Fig.3d: Please show 0 axes, and highlight dot corresponding to example neuron shown in Fig.2.

I really enjoyed reading your paper.

Máté Lengyel

Reviewer #3:

Remarks to the Author:

The paper by Hénaff et al. addresses the long-standing problem of how a population of neurons represents not just the most-likely stimulus, but also the uncertainty in that estimate. They combine electrophysiology in two visual cortical areas with two population-level models to argue that gain variability in individual neurons is effectively tuned for uncertainty.

The analyses are interesting, thoughtful and comprehensive, and the figures do a wonderful job of developing the story. That said, I have three major conceptual concerns that I believe should be clarified, after which I believe the paper is suitable for publication. I have tried to suggest how these concerns might be alleviated, but solutions to #1 and #3 aren't completely clear to me.

1) Precise representations of noisy stimuli are different from noisy representations of precise stimuli. Much of the writing in the paper suggests that the role of V1 neurons is to represent only the center-orientation of a stimulus distribution (rather than representing the entirety of any stimulus distribution). Therefore, the fact that a neuron's tuning curve is broader or has lower Fisher information when the stimulus has a broad orientation-distribution does not inherently tell us anything about the reliability of that neuron, since we cannot distinguish a precise representation of a noisy stimulus, from a noisy representation of a precise stimulus.

Wording around this should be clarified, e.g. on p2, I disagree that "the firing rate of V1 complex cells reports the total amount of energy at a particular orientation". Rather, V1 cells are responding to all orientations visible at a particular time, and the responses further depend on interactions between those orientations (and earlier orientations, although adaptation is not relevant here). The authors hint at this issue later in the sentence "Indeed, increasing stimulus spread increases perceptual discrimination thresholds because it acts as external orientation noise." However, this is simply acknowledging the reality that perceptual variability could reflect stimulus variability or neural variability.

In addition to rephrasing these types of arguments (not just these two specific examples, but throughout the paper), I think some analysis needs to be performed that better illustrates how tuning curves and Fisher information (e.g. Fig 2bc) depend on the stimulus bandwidth. Essentially, the population isn't trying to represent the single, central orientation - it's trying to represent the entire stimulus. In that case, Fisher information may not be the best metric, since it's really concerned with discriminating two nearby orientations, which the authors' stimuli make difficult. In essence, Fisher information appears to be used to support the argument that a neuron can't reliably discriminate orientation because of its tuning curve slope or inter-trial variability, but in fact those two orientations might be indiscriminable because of the broad distribution of simultaneously-presented orientations.

2) Appropriateness of correlating data across multiple factors

In figures 2f and 3, data are shown comparing gain variability and orientation uncertainties when two factors (orientation spread and contrast) are manipulated. While reporting a correlation is fine, I'm cautious about readers then interpreting this as implying that stimulus spread and contrast are tweaking the same underlying mechanism. This should be discussed. A better analytical approach, rather than only presenting correlations, might be to fit a two-factor model to the data or to report a metric like intra-class correlation.

3) Equivalence between intra- and inter-neuron variability

The decoding model involves 5 subpopulations of 25 or 50 neurons. This seems to be a neat way of changing from analysing variability across trials (within a neuron) to variability across neurons (within a trial). However, some justification for this approach should be included.

It would be helpful if readers could understand how this variance-across-neurons might be implemented in a more realistic, heterogeneous population. Single-trial estimates of variance only seem available here because neurons have exactly the same mean tuning curve. In a real population, if variance across neurons was calculated for a pool of similarly-tuned neurons, it would be artificially elevated because of slight differences in the tuning properties of each neuron in the pool. From the decoder's perspective, it's then unclear where that variance has arisen – from a noisy stimulus, or a noisy set of neurons. Conceptually, this is similar to the problem raised in point #1 above, where neural response variance across trials arises because of stimulus variance.

Minor

Compounding the issues in point 1, above, an additional source of neural noise is the variable temporal frequency for each stimulus component. The contribution of this should be discussed, along with the rationale for including TF variability.

In Fig. 3c, is it $n=78$ or 79 like the other panels?

In Fig 4, can the authors provide any intuition for why the null distribution is not centered on 0? Is this simply a result of the asymmetry of logarithms combined with a floor-effect for gain variability at 1?

In Fig 5c, are we seeing the mean model fit to individual neurons, or simply the fit to the mean of all neurons. It would be helpful to know the distribution of goodness of fit across the population.

It's not clear why the number of neurons in the decoder changes between Fig 7 & 8.

What might be the contribution of structure noise correlations on the decoding?

Response to reviewers

We thank the reviewers for their constructive feedback. Since most of their questions and comments were fairly diverse, we simply address them in order. We report the reviewers' comment in italic and include our point-by-point responses in regular font. We have combined the responses to several reviewers' comments under section headers which we refer to in the point-by-point responses. Finally, we have highlighted all changes to the original manuscript in bold.

Reviewer #1 (section 1)

This manuscript suggests to provide a new theory of the representation of uncertainty in the visual cortex. According to this view, "... average neural response strength encodes stimulus features, while cross-neuron variability in response gain encodes the uncertainty of these features...". To support their claims, the authors analyse anesthetized macaque single cell recording from V1 and V2, and make several observations based on their proposed model that handles both mean values and uncertainty of the neural signal.

While the topic of the paper is worthwhile, I have several problems with the logic and the delivery of the ideas presented in the paper. I am going to list these by contrasting the manuscript to the paper of Orban et al. (2016, Neuron), first since the Orban et al. paper has the most comprehensive treatment of the issue of variability and uncertainty for neural coding to date, and second because the authors themselves compare their work to that paper.

The original work (Goris et al, 2014, Nat Neu,) which serves as a conceptual basis of the present paper introduced the idea of a gain signal G , which is independent of the sensory input, and as such, brings in an extra term of variability. With an appropriate fitting of the parameters of this extra term, the joint variability of this "modulated Poisson" process could capture the supralinear nature of the variance-to-mean relation found in anesthetized recordings. This modulated Poisson model is a generic descriptive model that does not specify – beyond some general terms, such as "attention" - the origin or the fine structure of gain variability other than pegging it to a gamma distribution.

In the present manuscript, the authors move forward and look for the source of the gain signal and its variability. They embrace the concept that mean firing rates and variability of cell responses are separate and independent measures related to environmental features and uncertainty in encoding those features, respectively. However, the authors seem to embrace this idea a bit too strongly. Throughout the manuscript, it feels as they imply that this is an original contribution of the paper. For example, the starting line of the discussion says: "We have introduced a new perspective on the neural code. In our view, sensory neurons report the features of the environment and the reliability of this message through two different communication channels: the mean spike count and its variance. For example, a change in stimulus orientation might alter the mean firing rate of a V1 neuron, but it will not change its gain variability."

In fact, the concept that "average neural response strength encodes stimulus features, while cross-neuron variability in response gain encodes the uncertainty of these features" (c.f. abstract of this manuscript) has been prominently proposed in Orban et al. (2016) together with a particular implementation of this concept through neural sampling, and the proposed model was extensively tested against a large set of data including the example above with the orientation-invariance of response variability. Thus, the contribution of the present manuscript is not introducing the new perspective of separable coding of mean values and uncertainty, but an implementational trick, postulating an "uncertainty receptive field" in order to encoding some measure of uncertainty separately. While Orban et al. (2016) used sampling for this based on the concept of coding the joint posterior, the present paper uses a PPC-like approach with doubling the system with positing a second set of RF structure. It is fine to explore this implementation and link the gain concept to uncertainty this way, but the entire manuscript should do a much better job clarifying what is their own contribution and what are the ideas incorporated from previous research, especially when the given research has been referred to in the manuscript.

Author response: Relationship to other theories of uncertainty

Reviewers #1 and #2 asked us to clarify the relationship between our framework and two alternative theories of the neural representation of uncertainty: probabilistic population codes (PPC), and the sampling hypothesis. In particular, reviewer #2 identified three axes along which these theories differ, and asked us to position our results with respect to them. We agree that this is a fruitful discussion, and include it in our revised manuscript. The first axis concerns the use of response variance to encode stimulus uncertainty. We agree with the reviewer that our theory and results are aligned with the sampling hypothesis in this respect. The second axis pertains to whether uncertainty is encoded in the patterns of spikes across time or across neurons. In this aspect, our theory is aligned with PPC in that gain variability is too slow to be analyzed over time and can only be read off from inter-neuronal patterns of activity. While it is true that variants of the sampling hypothesis which use population activity have been proposed, our results contradict those that do not. Furthermore, comments from reviewer #3 indicate that it is not self-evident to the community that these two variants are equivalent. Finally, the third axis contains several aspects. While we agree

with the reviewer that distinguishing between parametric and non-parametric forms of inference is not helpful here, there remains an interesting distinction between feedforward inference (as in PPC) and recurrent inference that is often used in the sampling hypothesis. When not using numerical integration (Orban, 2016), most sampling-based models use recurrent computations to arrive at samples from the posterior (Buesing et al, 2011; Hennequin et al, 2014; Aitchison et al, 2016; Echeveste, 2019). In contrast, in our framework the first volley of spikes directly encodes a stable distribution over the environment, aligning it with PPC. In addition to this conceptual difference (which may have behavioral implications), this distinction comes with practical benefits, notably when decoding or fitting the model to data. In Orban et al's 2016 model, evaluating the likelihood of a stimulus requires a high-dimensional integration via Monte Carlo sampling. In our framework it is available in closed form, allowing us to straightforwardly fit the uncertainty receptive field to spike-count data (Fig. 5c). As such, our theory benefits from both the explanatory power of the sampling hypothesis and the simplicity of PPC.

Nevertheless we have modified the description of our work in three locations (abstract, introduction, and discussion) to clarify the nature of our contribution.

Reviewer #1 (section 2)

Once the truly new ideas of the manuscript are identified, the next question is to see the solidity and contributions of these ideas that advance our understanding beyond what was known before. Here, I have the following questions.

Regarding the concept of the uncertainty RF (URF), it is unclear what this URF really represents. The descriptive model of Goris et al (2014) partitioned variability in a mechanical sense: some variability comes from the sensory input and some other comes from other sources (top-down effects, local connection, etc.). This is fine for approximating measured variability, but tying this model conceptually to uncertainty is problematic. Should we take the authors statement "We propose that, while average response magnitude encodes stimulus features, variability in response gain encodes the uncertainty of these features. (p. 1)" literally? In other words, is it only the part of variability delivered by the gain term that defines uncertainty in the system and the point-process variability does not matter? If so what does this measure represent? It cannot represent the cell's full uncertainty about the interpretation of the stimulus since that would clearly require considering the quality of the input information. If the argument is that momentary gain variability essentially incorporates input variability of the previous moment (as suggested by the authors' wording e.g. p.2), then the separation into gain variability and input variability is unnecessary. In this case, the bona fide measured variability of the cell response that counts, the concept of URF is unnecessary, and the proposition of the manuscript is very much the same as that of Orban et al. (2016): a measurement of variability conveys information about uncertainty. The results and contributions of the manuscript in this case can be summarized as this: taking the restricted conditions of anesthetized animals and orientation selectivity, when neural spiking behaviour is modelled by the Goris et al (2014) model, the variability of the Gain term closely follows the true uncertainty of the cell about the stimulus, and so a hypothesised summary measurement of this variability, i. e. the width of the URF gives similar results to other measurements using a more comprehensive measure of the full variability. Under these conditions, variability can also be approximated with simple divisive normalization.

If this characterization is correct, two important questions are whether the present model has distinguishable behaviour from other models and whether that distinguishing behaviour provides additional benefits in terms of deeper understanding of neural coding. Since, it is unclear how the measure of variability conveyed by the URF compares to other measures of variability in the cell response, it would be very beneficial, if the authors could clarify this and list features that set apart their model from others.

Author response: Comparing measures of response dispersion

Reviewer #1 asked how gain variability compared to other measures of neuronal variability. We therefore also examined the relationship between stimulus uncertainty and Fano factor, an often-used measure of response dispersion. In contrast to gain variability, which captures a stable property of a neuron, this statistic depends on stimulus drive and count window. To obtain a single value of Fano factor for each stimulus family, we computed an estimate for each stimulus condition and then averaged these estimates across all stimulus orientations. This analysis revealed that the positive association of gain variability with stimulus uncertainty is somewhat unique. Specifically, Fano factor is not associated with stimulus uncertainty for either of our experimental manipulations (Fig. 3d; Fig. 4e).

Some studies have considered other measures of response dispersion, such as mean-matched Fano factor, spike-time reliability, and coefficient of variation. We opted to limit our comparison to Fano factor since this is arguably the most popular measure of response dispersion, and since it could in principle be used by the brain to represent uncertainty. Mean-matched Fano factor and spike-time reliability cannot be estimated instantaneously, and thus seem improbable candidate statistics to represent uncertainty.

Reviewer #1 (section 3)

If, as it was presented in the discussion, the authors suggest that their "width of the distribution" parameter is a quick-and-dirty, but efficient feed-forward first approximation of the ongoing true uncertainty, it would be very nice to spell out the range of its applicability. Specifically, under what conditions is this mode sufficient? Is it useable in natural situations? Is it necessary to

assume this separate mechanism for the feed-forward pass which is augmented by other uncertainty representations (c.f. active vs. passive comparison in Discussion p. 5), or it can be viewed as re-describing the initial segment of the more complete uncertainty estimation process used in active vision with complex natural conditions? If these are separate mechanisms, how do they work together; is there a continuous combination of information, a hand-shaking/take-over between the two mechanisms, if so, when and how? Answering these questions would greatly enhance the impact of the paper.

Author response: Generality of the uncertainty receptive field coding scheme

Reviewer #1 asked under which conditions the approximate inference scheme we propose is sufficient, and when it may need to be refined with further processing. Indeed, as we consider stimuli of increasing complexity, the distribution over features captured by a given stage in the hierarchy will also become more complex. For example, plaids and textures could present multi-modal distributions over orientation energy, and the simple coding scheme we propose may no longer suffice for orientation selective neurons in V1. Downstream areas, which encode more abstract and invariant features may be able to capture this distribution with a single mode however. Regardless, both in the case of complex stimuli and in the context of a task, V1 neurons would benefit from feedback and recurrent connections to refine their estimates of the environmental features and associated uncertainty (Haefner, 2016). We acknowledge this point in the discussion to point to limitations of our coding scheme.

Reviewer #1 (section 4)

One benefit would be if the model could be extended beyond the tested data set. From this point of view, Orban et al. (2016) delivered an analysis showing how some aspects of the structure of variability can be tested in anesthetized animals while other cannot. In the same way as classical receptive fields have diminishing descriptive power in awake animals performing a complex task, it is a question how well the proposed URF scheme hold up in such conditions. A test with such a data set would be illuminating.

Author response: Role of anesthesia

Reviewers #1 and #2 wondered whether the effects we reported might in part be due to the consequences of anesthesia. We are currently conducting follow-up experiments in awake fixating macaques that, in our opinion, alleviate this concern. In these experiments, we use the same stimuli as in the acute study (but only 2 rather than 5 different levels of spread), and record neural activity from the primary visual cortex using multi-laminar electrode arrays. We have included a figure in this response letter which shows that all of the main effects we report in our manuscript (including the positive relationship between gain variability and stimulus uncertainty, and slow gain-dynamics) generalize to the awake brain (Fig. 1). We are happy to share this analysis with the reviewers in the context of a discussion about the role of anesthesia. However, the goal of these follow-up experiments is to push our framework further and investigate how to decode stimulus uncertainty on a single-trial basis from simultaneously-recorded population responses. For this reason, we would like to refrain from publishing these new data in the current manuscript. We hope the reviewers will understand this.

Reviewer #1 (section 5)

A prominent claim of the manuscript is that the benefit of the URF formalism is easy decodability. In the Orban et al (2016) paper, an analysis was provided showing that measuring orientation with the nuisance parameter of contrast is special in the sense that this is almost the only setup where linear decoding could operate close to optimal (cf. Suppl Mat). Selecting any other combination of feature/nuisance e.g. feature= orientation and nuisance= phase or aperture, the linear decoding falls short of optimal decoding sometimes as much as by 40%. Can the present model be extended to nonlinear decoding and can uncertainty be computed efficiently in that situation?

Jozsef Fiser

Author response: Joint decoding of all stimulus features

We have generalized our decoding analysis to simultaneously infer stimulus contrast, orientation and spread (Fig. 7). In addition to simplifying the analysis and interpretation (reviewer #2), this shows that the decoder can seamlessly handle multiple nuisance parameters, including one which differs from contrast, stimulus spread (reviewer #1).

Reviewer #2 (section 1)

How do neural circuits represent uncertainty? This is one of the biggest open questions in systems neuroscience, and this paper does a beautiful job at starting to resolve this question in the macaque early visual cortex. In a nutshell, the authors find that it is the variability of neural responses that encodes uncertainty, this variability is well captured by variability in neural gain, and even though it can only be read out on long time scales from single neurons, it can still be read out efficiently on short time scales from a population of neurons. These are important findings on their own right, but also because they offer important support for one particular theory of the neural representation of uncertainty over alternative theories (see more on this below).

Overall, the paper is very well written with very informative (and highly aesthetic!) figures, thorough analyses and an impressive combination of theory and experiments. I am confident that it will be a must-read in the field. Nevertheless, I would like the authors to address some important points in the final version.

1. It is precisely because of the predicted high status of this paper in the field that I think the authors have a special responsibility of interpreting their results carefully in the context of the literature on neural representations of uncertainty, in particular the two competing theories that they single out: PPCs and sampling. [Disclaimer: as the authors will have guessed, this reviewer is a proponent of the latter — see my signature below.] While the Introduction clearly states that the paper offers a concrete example of sampling ("Here we incorporate this hypothesis [of Ref. 9 — a key paper on sampling] in the canonical model of neural coding"), the Discussion tries to give a more "balanced" view, trying to keep equal distance from the two camps. While being balanced is the right thing to aim for, here I think the authors reach a somewhat false balance, based on (and perpetuating) a misunderstanding of the main difference(s) between PPCs and sampling. As Ref. 9 argues, the main difference is that in PPCs (and some other representations) "both the mean of a probability distribution and the associated uncertainty are represented by time-average neural responses", while in sampling "the mean and variability of responses [serve] as independent information channels, respectively encoding the mean and the associated uncertainty of the probability distribution over visual feature". Compare this to how the authors of this paper characterise the main point of their paper (second sentence of the Discussion): "In our view, sensory neurons report the features of the environment and the reliability of this message through two different communication channels: the mean spike count and its variance." I think, based on this, it is pretty clear that the author's results fall squarely within the realm of sampling.

It is true that there is a secondary difference between PPCs and sampling: that PPCs are "instantaneous" (inasmuch as mean responses can be quickly read off), while sampling is usually thought of as a process evolving through time. However, it is a general (and by now mostly self-evident) rule in neural coding (of about anything) that you can buy time by neurons (or vice versa, if neurons are the more limited commodity) by "averaging" (computing statistics) across neurons rather than time. Indeed, in the specific context of sampling, Ref. 9 explicitly mentions this possibility: "Although, by using over-complete representations, in which many neurons effectively code for the same variable [which is precisely the case the authors explore in this paper in Figs. 7-8!], even one sample of a population activity pattern may represent multiple samples of the relevant variables, such that the effective rate of sampling can be faster than expected from neural time constants", and Savin and Deneve (cited in Ref. 9) provide a worked out example for this (again, in the context of sampling).

There is a third (seeming) difference between PPCs and sampling that the authors emphasise: that PPCs encode parametric distributions, while sampling represents "an arbitrarily flexible posterior distribution". However, strictly formally, samples can also be interpreted in a parametric form, as a mixture of deltas. More importantly, in the context of neural representations, the posteriors that need to be represented are themselves already considered to be a highly constrained parametric family (eg Gaussian in the Gaussian scale mixture model, as in Ref. 9), and once neural circuit mechanisms are proposed, these also obviously place severe constraints on the parametric form of distributions that can be represented (eg Buesing et al, PLoS Comput Biol 2011; Hennequin et al, NIPS 2014; Aitchison et al, PLoS Comput Biol 2016 [both Gaussian]; or Echeveste et al, bioRxiv 2019). (So, technically, in ML terms these circuits perform amortised variational inference while using samples to represent the resulting posteriors.) These parametric constraints also mean that should decoding (of uncertainty or else) be necessary, it can also be greatly simplified (and potentially sped up, see above) by taking them into account appropriately. Therefore, this distinction is largely a red herring, at least without a more nuanced discussion.

So, I have two (plus one) concrete suggestions for how to present this whole issue in the Discussion more fairly. First, I feel the first sentence of the Discussion is somewhat overstated by claiming a "new perspective on the neural code" which is supposed to refer to the idea of the "two different communication channels" described above. The same goes for the sentence later in the same paragraph stating that "this is a new conceptualization of the neural code". As the quote from Ref. 9 above shows, this idea as such is very much the same as that in Ref. 9 (and related papers) and the paper is strong enough without making this claim — so I would present this more in the way the Introduction does, saying that the paper presents (and tests!) a specific, novel instantiation of the sampling framework. Second, when discussing the PPC vs sampling distinction, and the results of this paper in the context of these, make clear that there are these two different "axes" along which they differ, with the second (time vs "space") being secondary to the first (same vs different information channels for communicating mean and uncertainty). I would either not mention the third axis (parametric vs "arbitrary" distributions) or with the important disclaimers mentioned above.

Author response: Please see the response entitled "Relationship to other theories of uncertainty" above.

2. In practical terms, given the arguments above, I have a couple of questions and suggestions for the decoding part.

- (a) Fig. 7b shows the orientation tuning curves of the simulated population. Please also state explicitly (and preferably show) how, if at all, mean responses depended on contrast.

Author response: We now detail explicitly how the mean responses depend on contrast, i.e. via the divisive normalization computation.

- (b) At present, only one decoder, an approximation of the optimal maximum likelihood decoder, is studied. In order to establish how important the “second information channel” of variability (see above) is, it would be useful to construct a decoder that only uses the mean responses and quantify how it fares relative to the current one that makes use of response variability. In addition, along the lines of Ref. 9, a proper comparison with a (linear) PPC (to see how important contrast- or other “nuisance” parameter-dependent modulations of variability — ignored by PPCs — are) requires training a linear decoder at one contrast (nuisance) level (or with contrast unknown) and again comparing its performance with that of the current decoder.

Author response: Reviewer #2 asked how our decoder would compare with a decoder that doesn't rely on variability and with a linear PPC trained for one contrast and spread. While we consider these to be interesting questions, we feel they are outside the scope of our paper. We argue here that gain variability is a candidate currency of uncertainty because of the properties we have discovered. We also show how these properties could emerge from known neural computations, and how uncertainty could potentially be decoded from a neural population. Our argument is that this is a code that could be used by neural circuits, rather than one that *should* be used. Under which circumstances it performs better than other codes remains to be seen. It is an important question, but we feel that it should be dealt with in future work, and ideally be evaluated in the context of awake recordings, in which this performance can be directly compared to behavioral read-out.

- (c) Perhaps it's worth considering repeating these analyses with the other main modulator of uncertainty considered in the paper: stimulus spread.

Author response: Please see the response entitled "Joint decoding of all stimulus features" above.

3. I have two more relatively major comments regarding two central findings of the paper.

- (a) A central finding of the paper is that changes in above-Poisson variability are best characterised as changes in the response gain of neurons. However, these gain changes are inferred from patterns of variability, not observed directly. An alternative hypothesis is that above-Poisson variability is caused by fluctuations in firing rate which in turn are due to fluctuations in membrane potential, or the total input to cells (Hennequin et al, Neuron 2018), and uncertainty could be represented by these fluctuations instead (as in Ref. 9 or Echeveste et al, bioRxiv 2019). These alternatives might be distinguishable from the gain fluctuations-based mechanism proposed here, based on the precise scaling of spike count variance with mean spike count and with the measurement time window (ie the kind of analyses performed in the paper). Some quick back-of-the-envelope calculations based on Hennequin et al (arXiv, 2014) suggest that a standard rectified power law nonlinearity mapping from membrane potentials to instantaneous firing rates, with the appropriate choice of the power law exponent and limit in the time scale of fluctuations (fast vs slow), may be able to result in the right scaling of spike count variability with either factor but perhaps not both simultaneously. In any case, this potential alternative should at the very least be discussed, if not analysed properly.

Author response: Reviewer #2 pointed out that gain fluctuations are not observed directly and that we cannot identify their mechanistic basis. We fully agree and have added a few sentences to the discussion acknowledging this fact. Our model offers a statistical description of neural activity, not a mechanistic one.

- (b) Another key finding is that gain fluctuations are slow(er than 1 s). I think some discussion would be due here as to how much this may be specific to the anaesthetised case, where slow fluctuations are known to dominate which are in turn largely absent in the awake V1 (Ref. 29, but to some degree also Ref. 15 for MT by the senior author).

Author response: Please see the response entitled "Role of anesthesia" above.

4. Minor comments

- p.1, Results, para 1: “The optimal estimation strategy” I think some qualification is needed here as maximum likelihood described here is not universally optimal.

Author response: We have replaced our references to "the optimal decoder" with a more nuanced description of maximum-likelihood estimation.

- p.3, para 1 of “The uncertainty receptive field as a consequence of stochastic divisive normalization”: “Finally, divisive normalization can be instantiated in image-computable models” Please expand on what you mean by “image-computable models”.

Author response: We clarified the meaning of ‘image-computable’ models.

- p.3, last para of “The uncertainty receptive field as a consequence of stochastic divisive normalization”: “Specifically, we fit the only free parameter” I assume that this means that beta and p have been separately fitted to the means, that’s why at this point σ_N is the only free parameter left. Please explain this explicitly.

Author response: We clarified our sequential fitting procedure of the uncertainty receptive field in the main text and in the methods section.

- p.9, para 1: “we synthesized 1,000 data-sets imposing slow gain dynamics, and 1,000 datasets imposing fast gain dynamics” Please specify what time scales were used exactly in the two cases, and what was the functional form of the dynamics (e.g. AR(1)?). Instead of performing model selection between two extreme cases (“slow” and “fast”), could you have fitted quantitatively the time scale of dynamics? Please explain your rationale.

Author response: When performing the recovery analysis for our assessment of gain dynamics, we synthesize datasets whose gain is piecewise-constant. For "slow dynamics" the gain is constant for the entire 1,000 ms; for "fast dynamics" it is constant over 62.5 ms windows (reviewer #2).

Author response: Reviewer #2 asked why we limit our analysis of gain dynamics to two extreme timescales. In our view, these timescales (a very fast vs very slow process) represent the most interesting scientific hypotheses, as they either enable a time-based code, or necessitate a population code. The fast timescale is based on the shortest time-window that is supported by our data (any shorter and it would be difficult to estimate empirical mean and variance) and still relatively "slow" compared to neuronal time-constants. The fact that the analysis clearly identifies the slowest possible timescale (the full stimulus interval) as the better description suggests that a time-based code is very unlikely. In a more practical sense, the predictions of intermediate timescales are difficult to test in our analysis paradigm as it is not obvious how to align the counting windows with the occurrence of a new gain sample for the non-extreme cases.

- p.9, eq. for $\text{Std}[\lambda_i]$: There seems to be a factor of p missing from the RHS.

Author response: That is now fixed.

- p.9, para after eq. for σ_G : “fitting the neurons’ mean responses with a divisive normalization model ($p = 2$, $\beta = 0.64$)” How come you got precisely an integer value for p? Is this because you only allowed integers? Please explain.

Author response: The mean estimate of the exponent in the stochastic normalization model is 2.001, which we rounded to 2 (reviewer #2).

- p.9, para above last eq.: “given a collection of $n*m$ spike counts” Please use proper multiplication sign instead of asterisk.

Author response: Done.

- p.9, last eq.: Shouldn’t there be a delta t term multiplying $f_i(S)$ whenever it appears?

Author response: That is now fixed.

- p.10, description of the heuristic estimator: Perhaps it’s worth noting that maximising the (marginal) likelihood of σ_G and then maximising the likelihood of S having fixed σ_G at this value does not generally give the same solution as jointly maximising the likelihood wrt. both parameters simultaneously — even if these maxima can be computed exactly.

Author response: Please see the response entitled "Joint decoding of all stimulus features" above.

- p.11, caption of Fig.1: The “Hi Olivier” text seems unintended.

Author response: That is now fixed.

- Fig.2b: Please also show example tuning curve when stimulus spread is changed.

Author response: We now show additional tuning curves in figure 2b.

- Fig.3d: Please show 0 axes, and highlight dot corresponding to example neuron shown in Fig.2.

Author response: We now show the location of the example neuron in figure 3e (reviewer #2).

I really enjoyed reading your paper.
Máté Lengyel

Reviewer #3 (section 1)

The paper by Hénaff et al. addresses the long-standing problem of how a population of neurons represents not just the most-likely stimulus, but also the uncertainty in that estimate. They combine electrophysiology in two visual cortical areas with two population-level models to argue that gain variability in individual neurons is effectively tuned for uncertainty.

The analyses are interesting, thoughtful and comprehensive, and the figures do a wonderful job of developing the story. That said, I have three major conceptual concerns that I believe should be clarified, after which I believe the paper is suitable for publication. I have tried to suggest how these concerns might be alleviated, but solutions to #1 and #3 aren't completely clear to me.

1. *Precise representations of noisy stimuli are different from noisy representations of precise stimuli. Much of the writing in the paper suggests that the role of V1 neurons is to represent only the center-orientation of a stimulus distribution (rather than representing the entirety of any stimulus distribution). Therefore, the fact that a neuron's tuning curve is broader or has lower Fisher information when the stimulus has a broad orientation-distribution does not inherently tell us anything about the reliability of that neuron, since we cannot distinguish a precise representation of a noisy stimulus, from a noisy representation of a precise stimulus.*

Wording around this should be clarified, e.g. on p2, I disagree that "the firing rate of V1 complex cells reports the total amount of energy at a particular orientation". Rather, V1 cells are responding to all orientations visible at a particular time, and the responses further depend on interactions between those orientations (and earlier orientations, although adaptation is not relevant here). The authors hint at this issue later in the sentence "Indeed, increasing stimulus spread increases perceptual discrimination thresholds because it acts as external orientation noise." However, this is simply acknowledging the reality that perceptual variability could reflect stimulus variability or neural variability. In addition to rephrasing these types of arguments (not just these two specific examples, but throughout the paper), I think some analysis needs to be performed that better illustrates how tuning curves and Fisher information (e.g. Fig 2bc) depend on the stimulus bandwidth. Essentially, the population isn't trying to represent the single, central orientation - it's trying to represent the entire stimulus. In that case, Fisher information may not be the best metric, since it's really concerned with discriminating two nearby orientations, which the authors' stimuli make difficult. In essence, Fisher information appears to be used to support the argument that a neuron can't reliably discriminate orientation because of its tuning curve slope or inter-trial variability, but in fact those two orientations might be indiscriminable because of the broad distribution of simultaneously-presented orientations.

Author response. Reviewer #3 requested we refine our wording throughout the paper. We acknowledge that some overly terse descriptions of V1 selectivity could lead to a misunderstanding. We are in complete agreement with the reviewer that it is misleading to consider V1 neurons as only being responsive to a single orientation. For this reason, we use Fisher information to estimate the neural discriminability of different orientations of an entire stimulus pattern, regardless of the complexity of that pattern (e.g., the number and diversity of oriented elements it contains). We have clarified statements throughout the manuscript regarding V1 cells reporting energy across a range of orientations. We further refer the reviewer to the cited Goris et al. (2015) paper which contains a comprehensive analysis of the effect of stimulus bandwidth on tuning curves and Fisher information using the same data set. More generally, the uncertainty framework we propose indeed does not distinguish between precise representations of noisy stimuli (highly orientation-dispersed stimuli) and noisy representations of precise stimuli (low contrast single-orientation stimuli). We view this invariance to the origin of uncertainty as a desired feature of a general theory of neural uncertainty representation.

2. *Appropriateness of correlating data across multiple factors. In figures 2f and 3, data are shown comparing gain variability and orientation uncertainties when two factors (orientation spread and contrast) are manipulated. While reporting a correlation is fine, I'm cautious about readers then interpreting this as implying that stimulus spread and contrast are tweaking the same underlying mechanism. This should be discussed. A better analytical approach, rather than only presenting correlations, might be to fit a two-factor model to the data or to report a metric like intra-class correlation.*

Author response: Reviewer #3 questioned the appropriateness of correlating data across multiple factors and requested an additional statistic as well. We now additionally report the outcome of an ANCOVA-analysis on the relation between gain variability and orientation uncertainty

3. *Equivalence between intra- and inter-neuron variability. The decoding model involves 5 subpopulations of 25 or 50 neurons. This seems to be a neat way of changing from analysing variability across trials (within a neuron) to variability across neurons (within a trial). However, some justification for this approach should be included.*

It would be helpful if readers could understand how this variance-across-neurons might be implemented in a more realistic, heterogeneous population. Single-trial estimates of variance only seem available here because neurons have exactly the

same mean tuning curve. In a real population, if variance across neurons was calculated for a pool of similarly-tuned neurons, it would be artificially elevated because of slight differences in the tuning properties of each neuron in the pool. From the decoder's perspective, it's then unclear where that variance has arisen – from a noisy stimulus, or a noisy set of neurons. Conceptually, this is similar to the problem raised in point #1 above, where neural response variance across trials arises because of stimulus variance.

Author response: Decoding from heterogeneous populations. Reviewer #3 asked for some explanation for the conditions under which we can substitute across-trial variability (as in our analyses) for across-neuron variability (as in our simulations), and whether organising neural populations into identically-tuned subpopulations is necessary for single-trial decoding. These are important points which we now clarify in the main text, figure captions, and the methods. Regarding the first point, if we assume different trials to be independent from each other, then statistics computed across trials can be thought of as statistics computed across identically tuned, independent neurons. In our decoding analysis however, we relax the assumption of identically tuned neurons. Provided that they fire independently from each other, our maximum likelihood decoder (Fig. 7) is able to handle entirely heterogeneous neural populations (and even the independence assumption can be relaxed, see Fig. 8c). It is only when we search for a neurally-plausible decoder (which approximates the maximum-likelihood one with simple operations) that we make the additional assumption of identically tuned subpopulations (see Fig. 9). This is mainly for convenience in the derivation of the neurally-plausible estimator, and we believe that it is not fully necessary as inter-neuronal variance due to differences in tuning could be estimated over many trials and factored out on a trial-by-trial basis. We leave the details of this more general case for future work.

4. Minor

- *Compounding the issues in point 1, above, an additional source of neural noise is the variable temporal frequency for each stimulus component. The contribution of this should be discussed, along with the rationale for including TF variability.*

Author response: We included TF variability to ensure that the spatial interference patterns created by summing differently oriented sine wave gratings would be sufficiently diverse within and across trials. We sought to characterize how visual cortex represents mixture stimuli in general, not a very specific subset with distinct spatial features. As the mixing manipulation is intended to act as noise source in this paper, we think it is no problem that TF variability contributes to neural response variability.

- *In Fig. 3c, is it $n=78$ or 79 like the other panels?*

Author response: In figure 3c, $n = 78$. One neuron only responded to two out of the ten stimulus families (example neuron 2 from the Goris et al (2015) paper) and hence did not enable a meaningful correlation estimate (reviewer #3).

- *In Fig 4, can the authors provide any intuition for why the null distribution is not centered on 0? Is this simply a result of the asymmetry of logarithms combined with a floor-effect for gain variability at 1?*

Author response: Reviewer #3 asked us to explain why the null distribution in Figure 4 is not centered on 0. Consistently with the reviewer's suggestion, we have added the following sentences to the Methods section: "Because gain variability is a positive-valued statistic, estimation error can introduce a bias that depends on the magnitude of the mean response. Consequently, the null distribution need not be centered at zero."

- *In Fig 5c, are we seeing the mean model fit to individual neurons, or simply the fit to the mean of all neurons. It would be helpful to know the distribution of goodness of fit across the population.*

Author response: Reviewer #3 asked whether we fit the stochastic normalization model to individual neurons and reported its mean estimate, or whether we fit it to the cross-neuron average response statistics. We did both. To determine the parameters that govern the mean response of each neuron, we fit a divisive normalization model to each individual cell's responses. This is described in detail in the Goris et al. (2015) paper. Here, we computed the cross-neuron average for each of the model's parameters (except for normalization noise) and we estimated normalization noise by fitting the cross-neuron averaged gain variability.

- *It's not clear why the number of neurons in the decoder changes between Fig 7 & 8.*

Author response: We have updated the decoding analysis such that the number of neurons is the same in figures 8 and 9 (reviewer #3).

- *What might be the contribution of structure noise correlations on the decoding?*

Author response: The analysis presented in Figure 8 in part speaks to this question. Here, we introduced noise correlations by means of one common source of gain modulation, shared across all neurons in the population. The

Figure 1 Representation of uncertainty in a V1 population during fixation. **(a)** Orientation tuning of 20 (out of 32) simultaneously recorded V1 units from an awake, fixating macaque. **(b)** Gain variability as a function of stimulus spread and contrast, averaged across two populations of V1 neurons (consisting of 32 and 45 units, respectively). **(c)** Orientation uncertainty as a function of gain variability, averaged across two populations of V1 neurons. **(d)** We fit two models to these data: one with slow gain dynamics, and one with fast gain dynamics. We measured goodness-of-fit by computing the log likelihood of the data under each model. The symbol shows the average difference in log likelihood under both models for two populations of V1 neurons.

effect of this at the level of firing rates is that noise correlations are maximal for identically tuned neurons, and decrease as the signal correlation decreases (see Goris et al (2014) and Ecker et al., 2014) – sometimes referred to as "limited range correlations". As we show, the decoding is robust for this particular kinds of noise correlation. We have not investigated alternative structures. We reserve this for future work.

Reviewers' Comments:

Reviewer #1:

Remarks to the Author:

Authors original replies denoted with **

I read with interest the authors' replies to my (and the other reviewers') comments and in some aspects, they did clarify the message of the manuscript. However, I found that some important issues were still not satisfactorily settled. The data analyses are by-and-large all clear and nicely done. Virtually all trouble I still have with the manuscript originate from two main problems I see with the present version. First, I feel the conceptual/theoretical importance of various features of the model proposed by the authors is still not clearly conveyed. Second, the wording of the paper of its contribution is still sounds a bit misleading at some points. Below, I will first give my general comments by reflecting on the first part of the authors' rebuttal, then I specify where I suggest some adjustments of the text.

General comments:

Rebuttal 1 (Relationship to other theories):

This is possible the most important issue, not because of the PPC-sampling dichotomy, which of course I cannot completely distant myself from, but because the fundamental benefit of this paper to the community would be if it clearly delineated the important and not important aspects of various probabilistic approaches and the contribution of the manuscript to those aspects. As I mentioned above, I think this is still not done at the desired level. The main reason is that while the three axes identified for comparing sampling and PPC in Reviewer 2's comments are now incorporated in the text, the fact that those axes are not equally important theoretically is lost in the manuscript. Specifically, the first axis captures a fundamental difference among various probabilistic model classes, whereas the other two are more related to early implementations of the two discussed model classes (PPP vs. sampling). Not clarifying this difference in importance of the axes hinders a deeper understanding of the contribution of the paper for readers in the field. Specifically:

**Reviewers #1 and #2 asked us to clarify the relationship between our framework and two alternative theories of the neural representation of uncertainty: probabilistic population codes (PPC), and the sampling hypothesis. In particular, reviewer #2 identified three axes along which these theories differ, and asked us to position our results with respect to them. We agree that this is a fruitful discussion, and include it in our revised manuscript. The first axis concerns the use of response variance to encode stimulus uncertainty. We agree with the reviewer that our theory and results are aligned with the sampling hypothesis in this respect.

1a) We have an agreement on this point.

**The second axis pertains to whether uncertainty is encoded in the patterns of spikes across time or across neurons. In this aspect, our theory is aligned with PPC in that gain variability is too slow to be analyzed over time and can only be read off from inter-neuronal patterns of activity. While it is true that variants of the sampling hypothesis which use population activity have been proposed, our results contradict those that do not.

1b) As Orban et al (2016) spelled it out very clearly, the fundamental difference between the concept of PPC and sampling lies in the first point above: Mean and variance of neural responses independent channels from the standpoint of brain's representational activity (sampling) vs. not (PPC). The ratio of time- vs. space-related coding (i.e. how many neurons represent jointly the variability between one

and many) is secondary to the first point. Different schemes can be envisioned which use more or less "time-related" (I just use this expression as a short-hand) coding not affecting the fact whether they encode uncertainty and values independently (as was proposed in the original sampling framework) or not (as it is done in the original PPC model).

As the authors acknowledge, a number of variants exist for both sampling and PPC, and while their model indeed contradict those sampling model proposals, which use a single-neuron-variability, this contradiction indicates just a step forward in clarifying potential implementation schemes for the main concept the sampling framework promotes (i.e. independence of coding mean and variance of the probability distribution, c.f. axis 1) not a contradiction in concept. In contrast, any of the PPC models that tie mean activity and variability together are conceptually contradicting the model the authors propose. As such, while the authors are right in describing more similarities to sampling or PPC along these axes, it is misleading to play the "on the one hand, on the other hand" argument here, since the two hands are not at the same level of importance. The manuscript should clarify this point.

BTW, it would be very interesting to explore what modification of the classical PPC model would be required to integrate the implementation of inter-neuronal patterns to represent variability independently from the mean. However, the key concept in this exploration would still remain to achieve "independent representation of uncertainty through response variance", which requires a fundamental departure from the present definition of the PPC model to align it with what sampling models represent today. For this reason, I see the new model proposed by the authors exactly a nice attempt to make such modifications.

**Furthermore, comments from reviewer #3 indicate that it is not self-evident to the community that these two variants are equivalent.

1c) Indeed. This is why it is very important to clearly describe the the role of the present model in the landscape of existing models.

**Finally, the third axis contains several aspects. While we agree with the reviewer that distinguishing between parametric and non-parametric forms of inference is not helpful here, there remains an interesting distinction between feedforward inference (as in PPC) and recurrent inference that is often used in the sampling hypothesis. When not using numerical integration (Orban, 2016), most sampling-based models use recurrent computations to arrive at samples from the posterior (Buesing et al, 2011; Hennequin et al, 2014; Aitchison et al, 2016; Echeveste, 2019). In contrast, in our framework the first volley of spikes directly encodes a stable distribution over the environment, aligning it with PPC.

1d) First, parametric vs. non-parametric and feed forward vs. recurrent are different questions, so it is a shifting target talking about one under the label of the other one, which obscures the discussion. The authors agree with the "red herring" claim about the discussion of the parametric/non-parametric axis, this part of the discussion needs little further elaboration. Second, focussing on the FF/RC issue, as the authors acknowledge below, in case of naturalistic setup with more complex stimuli and tasks, recurrent inference might come in play more strongly (c.f. answer to R#1 S3). Therefore, it is perfectly fine to say that the present data is well captured by the proposed model, but it is a research question to be tested in the future whether the same will be true under more complex conditions.

**In addition to this conceptual difference (which may have behavioral implications), this distinction comes with practical benefits, notably when decoding or fitting the model to data. In Orban et al's 2016 model, evaluating the likelihood of a stimulus requires a high-dimensional integration via Monte Carlo sampling. In our framework it is available in closed form, allowing us to straightforwardly fit the

uncertainty receptive field to spike-count data (Fig. 5c).

1e) This is a great feature of the proposed model. The only thing that I would strongly emphasize again is that this, as the authors mentioned, is a good computational/practical tool for researchers. In other words, a conceptual difference in how to compute uncertainty, not a conceptual difference in whether uncertainty should be computed independently.

**As such, our theory benefits from both the explanatory power of the sampling hypothesis and the simplicity of PPC.

1f) This is, indeed, the contribution of the paper.

In summary, my understanding of the paper is as I described in my original review: 1) The authors provided a new model in which they embraced the main conceptual feature of models proposed earlier by e.g. sampling-based models, namely the independent encoding of features and their uncertainty. 2) The authors provided a new way of computing this uncertainty, which is not time-based but instantaneous interneuronal-pattern based. 3) The authors showed that under various experimental conditions in anesthetized animals, the proposed model well captures the uncertainty in question.

I think this is plenty for a paper, and it is enough for standing without claiming more.

In contrast, here is what the authors did not do in my view. 1) They did not meld sampling and PPC conceptually, since the conceptual difference is the in/dependence of mean and variance representation, and that is impossible to combine. Some things are either independent or not. 2) The authors did not provide a new theory of representation, since the above mentioned independence of mean/variance is the theoretically new proposal and it was put forward by others before. The authors did provide a computationally new proposal of a practical assessment of uncertainty leaving the door open for future testing whether this is simply a good measure for researchers or a method that is a) viable under more complex conditions, b) used by the brain with distinctive consequences.

**Nevertheless we have modified the description of our work in three locations (abstract, introduction, and discussion) to clarify the nature of our contribution.

1g) I have to admit, I was a bit surprised when reading the first part of the new text. The modifications of the first two mentioned locations (abstract and intro) were a single word in the abstract (view model) and zero change in the intro. Given the relatively substantial objection in the previous review about the framing of the work in the intro, this seems a bit of a minimal effort.

Rebuttal 2 (Relationship of variance to different measures):

**Reviewer #1 asked how gain variability compared to other measures of neuronal variability. We therefore also examined the relationship between stimulus uncertainty and Fano factor, an often-used measure of response dispersion. In contrast to gain variability, which captures a stable property of a neuron, this statistic depends on stimulus drive and count window. To obtain a single value of Fano factor for each stimulus family, we computed an estimate for each stimulus condition and then averaged these estimates across all stimulus orientations. This analysis revealed that the positive association of gain variability with stimulus uncertainty is somewhat unique. Specifically, Fano factor is not associated with stimulus uncertainty for either of our experimental manipulations (Fig. 3d; Fig. 4e).

2a) I am a bit puzzled by these results, as it had been shown in previous papers that Fano-factors are to some extent related to the variability of neural responses even if in a particular manner. As the authors also pointed out, measurement of the Fano factor for the quantifying variability per se might be complicated due to the measure's sensitivity to various parameters, and the use of anaesthesia brings in an extra element of complication so that these require a delicate treatment of the Fano factor when variability is under scrutiny (e.g. White et al. J. Neurophys. 2012). Could the authors explain why the Fano-factor is detached from stimulus uncertainty in their setup? Can it be because of the way they derived the single value for it? What does "somewhat unique" above mean? That gain variability is a more sensitive measure than Fano factor or that it is measuring a different thing?

Specific comments:

I) As I described above, I see the contribution of the paper to have a new measurement of uncertainty that helps treating probabilistic models, which are based on the previously proposed theoretical view of separable mean and variance coding in neural signal. In this light, either please, do not use the word "our theory" in a confusable manner (in the abstract and elsewhere) or else clarify explicitly that by "our theory" you specifically refer to the approximation of stimulus uncertainty by inter-neural patterns and not the previously published concept of the mean and variability of cell responses being separate and independent coding channels linked to features of stimuli and uncertainty about them.

II) P.6 last para: Please, provide a bit more info (even if in the Methods) about the exact calculation of the Fano factor, and give a reasoning as to why they do not reflect stimulus uncertainty in your setup.

III) P.9 para 4-5: Please, rephrase the paragraphs to clarify the points discussed in Rebuttal 1 above.

IV) Please, provide some explicit predictions that would be uniquely support the proposed coding mechanisms over alternatives.

Jozsef Fiser

Reviewer #2:

Remarks to the Author:

I thank the authors their responses and the changes they have made to the manuscript.

While the authors addressed most of my concerns, there are still a couple of outstanding issues that I would like them to settle.

1. Major issue: distinguishing between different probabilistic representations

I appreciate that the authors have taken on board the "axes" I suggested along which different probabilistic representations can be distinguished. However, I still take issue with how they represent these different axes as if they were of equal importance. Let me give you very concrete examples why I don't think this is the case if I take your results and criteria seriously, and why I'd suggest you refine the corresponding section of the Discussion accordingly.

The authors argue (both in their response and in the manuscript) that if a code can be decoded based on the activity of a population over some short (~100 ms) time window then it cannot be a sampling-

based code and must be a PPC-like population code. However, note that Orban et al (2016, Fig S5) also used a 20(!) ms time window to decode the stimulus from a population that (by construction) used a sampling-based representation and obtained reasonably high accuracy (with an optimal decoder, at high enough contrast) even at such an extremely short integration time (cf. the 100 ms minimum used here). (While Orban et al did not explicitly decode uncertainty, this would have given the same result following the author's approach here to estimate uncertainty based on the decoded stimulus — see also below.) Then, by the authors' own logic here (Discussion, para 3), this would mean that Orban et al's model did not employ a temporal but a population code. This is a concrete demonstration of the point I made in my previous review about the rather trivial exchangeability of time and cells, and clearly shows the futility of distinguishing between different probabilistic representations based on whether they can be decoded using this particular approach.

I also have problems with the way the contrast between recurrent and feed-forward computations is presented. Although the conceptual distinction in this case is clear but to say that divisive normalization is feedforward because there exist *phenomenological* models of which the algebraic form is such that it has no obvious "recurrent" terms seems like an overinterpretation of those models. In fact, the best mechanistic (neural network) models of divisive normalization in V1 e.g. by Ken Miller's group (the SSN) patently rely on recurrent interactions. If this is the case, then even the stochastic divisive normalization mechanism presented here relies on recurrent computations. Conversely, using a (stochastic extension of the) SSN model, Hennequin et al (2018, Fig.7) showed that these recurrent models act on quite fast time scales, <50ms, eg for the quenching of variability with stimulus onset (or contrast) — which, I add based on Orban et al (2016), is incidentally central to sampling-based representations. Indeed, Echeveste et al (2019, Figs. 6-7) then went on to show that sampling by recurrent interactions in the SSN with the right dynamical features (onset transients and oscillations) can be used to achieve 0.02 bits/neuron (1 bit in total for a population of 50 neurons) divergence from the target distribution within 100 ms. (Ironically, the "first volley of spikes" advocated by the authors here has a special role in achieving this precision with this speed there.) Again, this shows that the distinction between different probabilistic representations based on their requirements on time and recurrent interactions is at best secondary to that based on whether they use variability of responses as a separate information channel — and should be discussed as such.

Other issues of decreasing importance

1. Credit assignment for the two communication channels idea

The first sentence of the Discussion still very much reads as if the the core idea of the mean and variability of responses acting as the two separate communication channels would be novel in this paper. I think there are plenty of novel insights in this paper so that it does not need to claim this particular one — and the authors did not seem to contest in my previous review that this idea was already clearly and explicitly articulated and pursued by Orban et al (2016). So I suggest refining this sentence along the lines of: "We have proposed a model of sensory cortex in which neurons ..." → "We have proposed a model of sensory cortex based on the idea [REF 9] that neurons ...".

Similarly, on p.3, para 1: "we hypothesize that V1 neurons might encode reliability through a separate channel tuned to these features." → "we employ the hypothesis that V1 neurons might encode reliability through a separate channel tuned to these features [REF 9]."

2. Decoding

Uncertainty is decoded only through the stimulus being decoded. This is somewhat problematic: by this token, uncertainty is also encoded in the retina, as one can decode the stimulus from retinal

activity, too — this argument is analogous to the commonly used argument for why eg objects are not represented as such in the retina. (A more useful way of thinking about feature x of stimulus s being specifically encoded in responses r would be to require x to create an information bottleneck between s and r , so that r has maximal information about x but minimal information about s .) This issue should be discussed.

As an aside, the biological plausibility of the heuristic decoder employed at the end of this section seems like a long stretch. Just because some neural responses have been found to be reasonably well described by a *particular* combination of "sums, squares, and division" (p.9) it doesn't mean that *any* combination of these operations (and the particular one required by this decoder) can be implemented by neurons. Some disclaimer along these lines should be added.

Even more minor comments:

- Although I appreciate that uncertainty is estimated based solely on mean responses so that circularity is avoided when analysing how response variability is related to this uncertainty, it also comes with a cost of inconsistency: the Fisher information calculation is based on assumptions of simple Poisson variability (and conditional independence) that are inconsistent with the data (and it is this very systematic inconsistency that is central to the analysis of variability). This should be discussed, and I think the authors will agree that ultimately a (e.g. behavioural) measure of uncertainty that is altogether independent of the analysed neural responses should be used to identify its neural representation.

- p.8, column right, para 2: "V1 neurons whose responses resulted from linear filtering followed by stochastic divisive normalization" → As I understand from the Methods (see also below) you did not actually use your stochastic divisive normalization model to simulate responses, but instead used it simply to compute condition-dependent gain variances, and then used these to simulate responses from the modulated Poisson model (with a gamma distribution for gain that had the given variance). If my understanding is correct then the sentence quoted above is misleading and needs to be refined.

- Fig.8b, bottom panel, and c: I couldn't find how estimation variance is defined, please define it. As it seems to be measured in deg^2 it suggests that simply sample variance was used for a circular variable. If so, is there any reason why not a proper circular statistic of dispersion (eg mean resultant length) was used?

- page 6, penultimate para: "suggesting that they induce stimulus uncertainty for different reasons" → I didn't follow the logic here.

- *ibid.* "This suggests that gain variability represents the total amount of stimulus uncertainty, regardless of the source of this uncertainty" and p.7, end of para 1 "This invariance to the source of uncertainty..." , → But there are multiple cases when different conditions in Fig.3b correspond to the same level of uncertainty (y-axis) but have different amounts of gain variability (x-axis). Do these cases contradict these claims? Please discuss.

- There are multiple references in the decoding section (and associated methods and Fig.7 caption) to how the "dynamic range" of cells has been randomly chosen — but I couldn't find the (either formal or informal) definition of dynamic range anywhere.

- p.8, right column, end of para 1: "consistently with its interpretation as a consequence of stochastic normalization" → Why couldn't stochastic normalization take place on a short time scale in principle? Your specific stochastic normalization model does indeed have a slow time scale — but even that is

only because you choose epsilon to be fixed for the duration of a trial.

- p.14, para 1, 1st sentence: Give the functional form of g_i , and then explain further down where you talk about fitting mean responses how you fitted this form to data. And be more explicit about whether you fitted all neurons' data at once with a single set of parameters p and β , or you fitted each neuron's data separately and then averaged the best-fit parameter values across neurons. (I assume the "average estimate" expression implies the latter but better be sure.)

- p.14, last para of "Fitting the stochastic normalization model": "The stimulus-dependent normalization $\sum_j g_j(S)$ was computed by simulating responses of a fixed pool of neurons with a diverse set of tuning properties" → Give details about how this was done exactly, i.e. what were the set of $g_j(S)$ simulated. Also, did you assume the same pool for all neurons fitted?

- p.15, para 2: "We then applied the equations of the stochastic normalization model to obtain a firing rate and gain variability for each neuron." → What were the cells of the normalization pool in this case? Were these the same cells that were decoded, i.e. did you assume that you observe a complete population in which neurons divisively normalize one another? If so, shouldn't you look at how robust your decoding results are in the more realistic case if you only observe a subset of the population of neurons that provide divisive normalization to the neurons you are decoding (and when the divisive normalization pools of your decoded neurons only overlap partially)?

- p.15, after the first set of equations (btw, you should consider numbering your equations): say explicitly what $f_i(s)$ is in the equation for λ_i (I assume it's the same as in the stochastic normalization model).

- p.15, left column, last para: "with a random initialization" → If this means that you optimised the likelihood starting from a single initial condition, explain why you treated the likelihood as concave — otherwise you would have needed to try several initial conditions.

- p.15, interneuronal gain correlations: Once you create G by adding two gamma variates of different scale (because G_s and G_p are multiplied by different constants), I don't think it is gamma distributed any more — yet your inverse binomial decoding likelihood (implicitly) assumes a gamma distributed gain. Please comment on this. I also suggest using a symbol other than λ for the mixing ratio of shared and private gain variability to avoid confusion with λ_i used elsewhere (including the next paragraph).

- I would like to encourage you to consider sharing your data more openly than just "upon reasonable request".

- Ref 26 has been published (in J Neurosci) in the mean time.

Máté Lengyel

Reviewer #3:

Remarks to the Author:

This is a comprehensive response and the authors have dealt with all of my initial concerns.

Nicholas Price

Response to reviewers

We again thank the reviewers for their constructive feedback. We report the reviewers' comment in italic and include our point-by-point responses in regular font. We have combined the responses to several reviewers' comments in the initial section below. Finally, we have highlighted all changes to the updated manuscript in bold.

Author response: Relationship to other theories of uncertainty

In our previous rebuttal, we embraced the framework proposed by reviewer #2 for situating our model of the neural representation of uncertainty with respect to existing theories. In doing so we found that our proposal was similar to the sampling hypothesis in one respect (using response variability to encode uncertainty) but closer to linear PPCs in other respects (using variability across neurons rather than across time, and feed-forward rather than recurrent computation). In response to this, reviewers #1 and #2 now wish for us to downplay the importance of these latter aspects and emphasize the importance of the first aspect, a view that favors their own paper on the subject (Orban et al., 2016). Trying to assign the appropriate level of importance to each axis is inherently difficult, and perhaps not even fruitful. In our view, it belongs to the domain where scientific arguments are based on opinion rather than fact. Conversations with other researchers give us the impression that the view that only the first axis truly matters is not a widely held one. Moreover, some people have argued that even the first axis is not so clear cut as the reviewers claim. Specifically, our model can be framed as a nonlinear PPC (Xaq Pitkow, personal communication, see supplemental document for technical explanation). In our view, our proposal truly represents a bridge between the best-known instances of both model-classes. We wish to clearly acknowledge that the "variance as specialized uncertainty channel" idea originates from the sampling-hypothesis. As explained below, we have made several modifications to the manuscript to make sure there is no doubt about this. That said, we also want to make clear that we built on ideas which we feel are crucial contributions from the PPC models: axis 2 (cross-neuron patterns of activity) and axis 3 (feed-forward computation). In summary, it is our opinion that positioning our model along the three axes proposed by reviewer #2 places us at the intersection of sampling- and PPC-based models of uncertainty coding. We understand that some readers may feel that one of these axes has a privileged status, but we expect there will be a diversity of opinions about this issue. It is not our intention to escalate the debate with the reviewers. We hope they will grant us the room to express our opinion on the relation of our model to other models in our paper, even if that opinion differs from their own.

Reviewer #1: general comments

I read with interest the authors' replies to my (and the other reviewers') comments and in some aspects, they did clarify the message of the manuscript. However, I found that some important issues were still not satisfactorily settled. The data analyses are by-and-large all clear and nicely done. Virtually all trouble I still have with the manuscript originate from two main problems I see with the present version. First, I feel the conceptual/theoretical importance of various features of the model proposed by the authors is still not clearly conveyed. Second, the wording of the paper of its contribution is still sounds a bit misleading at some points. Below, I will first give my general comments by reflecting on the first part of the authors' rebuttal, then I specify where I suggest some adjustments of the text.

Reviewer #1: general comments, rebuttal 1 (Relationship to other theories)

This is possible the most important issue, not because of the PPC-sampling dichotomy, which of course I cannot completely distant myself from, but because the fundamental benefit of this paper to the community would be if it clearly delineated the important and not important aspects of various probabilistic approaches and the contribution of the manuscript to those aspects. As I mentioned above, I think this is still not done at the desired level. The main reason is that while the three axes identified for comparing sampling and PPC in Reviewer 2's comments are now incorporated in the text, the fact that those axes are not equally important theoretically is lost in the manuscript. Specifically, the first axis captures a fundamental difference among various probabilistic model classes, whereas the other two are more related to early implementations of the two discussed model classes (PPP vs. sampling). Not clarifying this difference in importance of the axes hinders a deeper understanding of the contribution of the paper for readers in the field. Specifically:

- ***Reviewers #1 and #2 asked us to clarify the relationship between our framework and two alternative theories of the neural representation of uncertainty: probabilistic population codes (PPC), and the sampling hypothesis. In particular, reviewer #2 identified three axes along which these theories differ, and asked us to position our results with respect to them. We agree that this is a fruitful discussion, and include it in our revised manuscript. The first axis concerns the use of response variance to encode stimulus uncertainty. We agree with the reviewer that our theory and results are aligned with the sampling hypothesis in this respect.*

1a) We have an agreement on this point.

Author response: Indeed.

- ***The second axis pertains to whether uncertainty is encoded in the patterns of spikes across time or across neurons. In this aspect, our theory is aligned with PPC in that gain variability is too slow to be analyzed over time and can only be read off from inter-neuronal patterns of activity. While it is true that variants of the sampling hypothesis which use population activity have been proposed, our results contradict those that do not.*

1b) As Orban et al (2016) spelled it out very clearly, the fundamental difference between the concept of PPC and sampling lies in the first point above: Mean and variance of neural responses independent channels from the standpoint of brain's representational activity (sampling) vs. not (PPC). The ratio of time- vs. space-related coding (i.e. how many neurons represent jointly the variability between one and many) is secondary to the first point. Different schemes can be envisioned which use more or less "time-related" (I just use this expression as a short-hand) coding not affecting the fact whether they encode uncertainty and values independently (as was proposed in the original sampling framework) or not (as it is done in the original PPC model).

As the authors acknowledge, a number of variants exist for both sampling and PPC, and while their model indeed contradict those sampling model proposals, which use a single-neuron-variability, this contradiction indicates just a step forward in clarifying potential implementation schemes for the main concept the sampling framework promotes (i.e. independence of coding mean and variance of the probability distribution, c.f. axis 1) not a contradiction in concept. In contrast, any of the PPC models that tie mean activity and variability together are conceptually contradicting the model the authors propose. As such, while the authors are right in describing more similarities to sampling or PPC along these axes, it is misleading to play the "on the one hand, on the other hand" argument here, since the two hands are not at the same level of importance. The manuscript should clarify this point.

Author response: Our goal is not to resolve the PPC vs. sampling debate, but rather expose the relevant axes on which these theories differ, and position our results with respect to them. Systematically exploring all three axes therefore helps us connect our model to and differentiate it from various specific instantiations of these theories. Furthermore, the different axes will likely be more or less relevant to different parts of the community: the use of response variance is interesting from an algorithmic viewpoint, but the use of inter-neuronal activity and feed-forward computation may be more so from a mechanistic one. We acknowledge (in the response and the current Discussion) that sampling-based models of uncertainty allow different implementations, and our contradiction of time-based schemes therefore provides a refinement of these theories. Similarly, note that PPCs also allow different implementations, and that our results specifically contradict their linear variant, not the concept as a whole.

BTW, it would be very interesting to explore what modification of the classical PPC model would be required to integrate the implementation of inter-neuronal patterns to represent variability independently from the mean. However, the key concept in this exploration would still remain to achieve "independent representation of uncertainty through response variance", which requires a fundamental departure from the present definition of the PPC model to align it with what sampling models represent today. For this reason, I see the new model proposed by the authors exactly a nice attempt to make such modifications.

Author response: We agree that this is a very interesting endeavor, and that our model represents a step forward in this direction. Whether our model is no longer a PPC because of its use of response variance seems like a debatable point which we would prefer to move beyond. As we mentioned above, the supplementary document we made available for the reviewers offers a technical explanation of nonlinear PPCs and argues that our results are consistent with this. This is not our own work (it is Xaq Pitkow's), and it is beyond the scope of our paper, but to the very least, it may help the reviewers appreciate that our findings do not necessarily contradict all PPC-models.

- ***Furthermore, comments from reviewer #3 indicate that it is not self-evident to the community that these two variants are equivalent.*

1c) Indeed. This is why it is very important to clearly describe the the role of the present model in the landscape of existing models.

Author response: As mentioned above, we have modified the discussion to express as clearly as possible which specific proposed models are consistent with our findings, and which are not.

- ***Finally, the third axis contains several aspects. While we agree with the reviewer that distinguishing between parametric and non-parametric forms of inference is not helpful here, there remains an interesting distinction between feedforward inference (as in PPC) and recurrent inference that is often used in the sampling hypothesis. When not using numerical integration (Orban, 2016), most sampling-based models use recurrent computations to arrive at samples from the posterior*

(Buesing et al, 2011; Hennequin et al, 2014; Aitchison et al, 2016; Echeveste, 2019). In contrast, in our framework the first volley of spikes directly encodes a stable distribution over the environment, aligning it with PPC.

1d) First, parametric vs. non-parametric and feed forward vs. recurrent are different questions, so it is a shifting target talking about one under the label of the other one, which obscures the discussion. The authors agree with the “red herring” claim about the discussion of the parametric/non-parametric axis, this part of the discussion needs little further elaboration. Second, focussing on the FF/RC issue, as the authors acknowledge below, in case of naturalistic setup with more complex stimuli and tasks, recurrent inference might come in play more strongly (c.f. answer to R#1 S3). Therefore, it is perfectly fine to say that the present data is well captured by the proposed model, but it is a research question to be tested in the future whether the same will be true under more complex conditions.

Author response: We agree (and say so in the discussion) that the feed-forward scheme we propose could be combined with recurrent ones, and that such an augmented scheme might be necessary to explain the representation of more complex forms of uncertainty.

- ***In addition to this conceptual difference (which may have behavioral implications), this distinction comes with practical benefits, notably when decoding or fitting the model to data. In Orban et al’s 2016 model, evaluating the likelihood of a stimulus requires a high-dimensional integration via Monte Carlo sampling. In our framework it is available in closed form, allowing us to straightforwardly fit the uncertainty receptive field to spike-count data (Fig. 5c).*

1e) This is a great feature of the proposed model. The only thing that I would strongly emphasize again is that this, as the authors mentioned, is a good computational/practical tool for researchers. In other words, a conceptual difference in how to compute uncertainty, not a conceptual difference in whether uncertainty should be computed independently.

Author response: Agreed. We think the discussion reflects this idea.

- ***As such, our theory benefits from both the explanatory power of the sampling hypothesis and the simplicity of PPC.*

1f) This is, indeed, the contribution of the paper.

In summary, my understanding of the paper is as I described in my original review: 1) The authors provided a new model in which they embraced the main conceptual feature of models proposed earlier by e.g. sampling-based models, namely the independent encoding of features and their uncertainty. 2) The authors provided a new way of computing this uncertainty, which is not time-based but instantaneous interneuronal-pattern based. 3) The authors showed that under various experimental conditions in anesthetized animals, the proposed model well captures the uncertainty in question.

I think this is plenty for a paper, and it is enough for standing without claiming more.

In contrast, here is what the authors did not do in my view. 1) They did not meld sampling and PPC conceptually, since the conceptual difference is the in/dependence of mean and variance representation, and that is impossible to combine. Some things are either independent or not. 2) The authors did not provide a new theory of representation, since the above mentioned independence of mean/variance is the theoretically new proposal and it was put forward by others before. The authors did provide a computationally new proposal of a practical assessment of uncertainty leaving the door open for future testing whether this is simply a good measure for researchers or a method that is a) viable under more complex conditions, b) used by the brain with distinctive consequences.

Author response: Regarding 1), we would agree with the reviewer if the use of response variance were the only distinguishing factor between sampling-based models and (linear) PPCs. If we also take into account the other aspects discussed above, then we do consider our proposal to be a hybrid of both existing theories. Regarding 2), we agree (and clarify in the text), that the idea of using response variance to represent uncertainty was already introduced by Orban et al. (2016). However, by introducing the notion of a parametric uncertainty receptive field which results from stochastic divisive normalization (thereby connecting these ideas to canonical descriptions of neural computation), we consider our model to indeed be a “new theory of representation”. Further tests of this theory, such as whether it can be directly connected to behavior, are indeed warranted. We hope the reviewer will agree with us that the issues on which we disagree (“Did we meld two different traditions –as we claim–, or is it impossible to meld these traditions –as reviewer 1 claims–?”; “Does our proposal constitute a new theory of representation or is that an overstatement?”) really belong to the domain of subjective opinion, and might even be largely semantic in nature.

- ***Nevertheless we have modified the description of our work in three locations (abstract, introduction, and discussion) to clarify the nature of our contribution.*

1g) I have to admit, I was a bit surprised when reading the first part of the new text. The modifications of the first two mentioned locations (abstract and intro) were a single word in the abstract (view model) and zero change in the intro.

Given the relatively substantial objection in the previous review about the framing of the work in the intro, this seems a bit of a minimal effort.

Author response: We have removed all references to "our theory" in the abstract, replacing them with references to "our model" instead. The introduction clearly references (Orban, 2016) when mentioning that response variability could represent uncertainty. Finally, we have re-written the initial sentences of the discussion which recapitulate our contributions.

Reviewer #1: general comments, rebuttal 2 (Relationship of variance to different measures)

***Reviewer #1 asked how gain variability compared to other measures of neuronal variability. We therefore also examined the relationship between stimulus uncertainty and Fano factor, an often-used measure of response dispersion. In contrast to gain variability, which captures a stable property of a neuron, this statistic depends on stimulus drive and count window. To obtain a single value of Fano factor for each stimulus family, we computed an estimate for each stimulus condition and then averaged these estimates across all stimulus orientations. This analysis revealed that the positive association of gain variability with stimulus uncertainty is somewhat unique. Specifically, Fano factor is not associated with stimulus uncertainty for either of our experimental manipulations (Fig. 3d; Fig. 4e).*

2a) I am a bit puzzled by these results, as it had been shown in previous papers that Fano-factors are to some extent related to the variability of neural responses even if in a particular manner. As the authors also pointed out, measurement of the Fano factor for the quantifying variability per se might be complicated due to the measure's sensitivity to various parameters, and the use of anaesthesia brings in an extra element of complication so that these require a delicate treatment of the Fano factor when variability is under scrutiny (e.g. White et al. J. Neurophys. 2012). Could the authors explain why the Fano-factor is detached from stimulus uncertainty in their setup? Can it be because of the way they derived the single value for it? What does "somewhat unique" above mean? That gain variability is a more sensitive measure than Fano factor or that it is measuring a different thing?

Author response: We have added a couple of sentences to the Results section and a supplementary figure to explain why Fano factor can be detached from stimulus uncertainty. "Somewhat unique" was simply meant to convey that gain variability is a more sensitive measure than Fano factor for stimulus uncertainty (the detachment can happen, but need not happen, it depends on the mean response).

Reviewer #1: specific comments

1. *As I described above, I see the contribution of the paper to have a new measurement of uncertainty that helps treating probabilistic models, which are based on the previously proposed theoretical view of separable mean and variance coding in neural signal. In this light, either please, do not use the word "our theory" in a confusable manner (in the abstract and elsewhere) or else clarify explicitly that by "our theory" you specifically refer to the approximation of stimulus uncertainty by inter-neural patterns and not the previously published concept of the mean and variability of cell responses being separate and independent coding channels linked to features of stimuli and uncertainty about them.*

Author response: In the abstract, we replaced references to "our theory" with references to "our model". In the introduction, the reference to "our theory" follows a detailed description of the model, hence we have left it as is. Similarly, in the first paragraph of the section "Testing the theory in visual cortex", the reference to "our theory" follows a description of the specific predictions of our model, hence we have left it as is. In the first paragraph of "Representation of uncertainty across the visual hierarchy", and the fourth and fifth paragraphs of the Discussion we replaced the reference to "our theory" with "our model".

2. *P.6 last para: Please, provide a bit more info (even if in the Methods) about the exact calculation of the Fano factor, and give a reasoning as to why they do not reflect stimulus uncertainty in your setup.*

Author response: Done.

3. *P.9 para 4-5: Please, rephrase the paragraphs to clarify the points discussed in Rebuttal 1 above.*

Author response: As we explained above, our opinion on this issue differs from the reviewer's opinion. We have modified these paragraphs slightly to more clearly express what we mean.

4. *Please, provide some explicit predictions that would be uniquely support the proposed coding mechanisms over alternatives.*

Author response: We have added a few sentences in the discussion that detail predictions regarding the relation between errors in perceptual orientation estimates and gain variability that could be investigated in an awake, behaving animal.

Reviewer #2 (Remarks to the Author):

I thank the authors their responses and the changes they have made to the manuscript.

While the authors addressed most of my concerns, there are still a couple of outstanding issues that I would like them to settle.

1. Major issue: distinguishing between different probabilistic representations

I appreciate that the authors have taken on board the "axes" I suggested along which different probabilistic representations can be distinguished. However, I still take issue with how they represent these different axes as if they were of equal importance. Let me give you very concrete examples why I don't think this is the case if I take your results and criteria seriously, and why I'd suggest you refine the corresponding section of the Discussion accordingly.

The authors argue (both in their response and in the manuscript) that if a code can be decoded based on the activity of a population over some short (100 ms) time window then it cannot be a sampling-based code and must be a PPC-like population code. However, note that Orban et al (2016, Fig S5) also used a 20(!) ms time window to decode the stimulus from a population that (by construction) used a sampling-based representation and obtained reasonably high accuracy (with an optimal decoder, at high enough contrast) even at such an extremely short integration time (cf. the 100 ms minimum used here). (While Orban et al did not explicitly decode uncertainty, this would have given the same result following the author's approach here to estimate uncertainty based on the decoded stimulus — see also below.) Then, by the authors' own logic here (Discussion, para 3), this would mean that Orban et al's model did not employ a temporal but a population code. This is a concrete demonstration of the point I made in my previous review about the rather trivial exchangeability of time and cells, and clearly shows the futility of distinguishing between different probabilistic representations based on whether they can be decoded using this particular approach.

Author response: The model used in Supplementary Figure 5 of Orban et al (2016) is indeed a spatial variant of the sampling hypothesis (42 neurons decoded from 20 ms of activity), a class of models which we acknowledge to be consistent with our results in both the Discussion and in our previous response. However, note that the model presented in the introduction and Figure 1 of Orban et al is explicitly time-based, with neural activity of individual neurons assumed to be independent after 20 ms. Our results contradict this assumption (gain fluctuations are better described as being slow than fast). That is the core of our logic: if gain fluctuations are slow, then time-based decoding is not a viable option. We think this is straightforward logic.

*I also have problems with the way the contrast between recurrent and feed-forward computations is presented. Although the conceptual distinction in this case is clear but to say that divisive normalization is feedforward because there exist *phenomenological* models of which the algebraic form is such that it has no obvious "recurrent" terms seems like an overinterpretation of those models. In fact, the best mechanistic (neural network) models of divisive normalization in V1 e.g. by Ken Miller's group (the SSN) patently rely on recurrent interactions. If this is the case, then even the stochastic divisive normalization mechanism presented here relies on recurrent computations. Conversely, using a (stochastic extension of the) SSN model, Hennequin et al (2018, Fig.7) showed that these recurrent models act on quite fast time scales, <50ms, eg for the quenching of variability with stimulus onset (or contrast) — which, I add based on Orban et al (2016), is incidentally central to sampling-based representations. Indeed, Echeveste et al (2019, Figs. 6-7) then went on to show that sampling by recurrent interactions in the SSN with the right dynamical features (onset transients and oscillations) can be used to achieve 0.02 bits/neuron (1 bit in total for a population of 50 neurons) divergence from the target distribution within 100 ms. (Ironically, the "first volley of spikes" advocated by the authors here has a special role in achieving this precision with this speed there.) Again, this shows that the distinction between different probabilistic representations based on their requirements on time and recurrent interactions is at best secondary to that based on whether they use variability of responses as a separate information channel — and should be discussed as such.*

Author response: We agree that divisive normalization is a functional description of neural activity which does not commit to a particular mechanistic implementation. In fact, there likely exist many different mechanistic implementations in different circuits and systems (Carandini and Heeger, 2012). As such, this functional model is able to describe both the steady-state of recurrent networks (such as the SSN), and purely-forward versions. Our point here is that the uncertainty receptive field does not *require* recurrent computation, in contrast to sampling-based models. This provides us with the distribution of neural activity given a stimulus in closed form (as is the case for PPCs). This has the immense practical benefit of being able to fit the model to neural data, and might also have computational benefits such as those variational auto-encoders (Kingma & Welling, 2013; Rezende et al, 2014) enjoy over recurrent sampling-based ones (Neal, 2012). We have modified the relevant part of the discussion to clarify our point.

Other issues of decreasing importance

1. Credit assignment for the two communication channels idea

The first sentence of the Discussion still very much reads as if the the core idea of the mean and variability of responses acting as the two separate communication channels would be novel in this paper. I think there are plenty of novel insights in this paper so that it does not need to claim this particular one — and the authors did not seem to contest in my previous review that this idea was already clearly and explicitly articulated and pursued by Orban et al (2016). So I suggest refining this sentence along the lines of: "We have proposed a model of sensory cortex in which neurons ..." → "We have proposed a model of sensory cortex based on the idea [REF 9] that neurons ...".

Author response: We agree that this is not the main contribution of the paper and have re-written that section to better describe the contributions of the paper.

Similarly, on p.3, para 1: "we hypothesize that V1 neurons might encode reliability through a separate channel tuned to these features." → "we employ the hypothesis that V1 neurons might encode reliability through a separate channel tuned to these features [REF 9]."

Author response: We have modified that sentence in line with the reviewer's suggestion.

2. Decoding

Uncertainty is decoded only through the stimulus being decoded. This is somewhat problematic: by this token, uncertainty is also encoded in the retina, as one can decode the stimulus from retinal activity, too — this argument is analogous to the commonly used argument for why eg objects are not represented as such in the retina. (A more useful way of thinking about feature x of stimulus s being specifically encoded in responses r would be to require x to create an information bottleneck between s and r , so that r has maximal information about x but minimal information about s .) This issue should be discussed.

Author response: Decoding uncertainty through the stimulus allows us to restrict our estimates to uncertainty values which can be generated by the uncertainty receptive field. This reduces the noise in our estimates of uncertainty. We further restrict these estimates by decoding uncertainty through the *latent parameters* of the stimulus (orientation, contrast, and spread). This allows us to leverage the knowledge that the observed patterns of activity do not arise from arbitrary stimuli, but instead a limited set of admissible ones.

Secondly, since the parameters of the stimulus (contrast and spread) are the only relevant factors governing orientation uncertainty (as opposed to the raw pixel values), decoding from a representation which exposes these parameters (such as our model V1 neurons) is preferable to one which instead encodes the pixel intensities. We consider this to be a fairly technical point that we prefer not to discuss in the paper.

*As an aside, the biological plausibility of the heuristic decoder employed at the end of this section seems like a long stretch. Just because some neural responses have been found to be reasonably well described by a *particular* combination of "sums, squares, and division" (p.9) it doesn't mean that *any* combination of these operations (and the particular one required by this decoder) can be implemented by neurons. Some disclaimer along these lines should be added.*

Author response: We have modified the main text's description of the heuristic estimator to make clear that its elementary operations are biologically plausible, not their particular combination.

Even more minor comments:

- *Although I appreciate that uncertainty is estimated based solely on mean responses so that circularity is avoided when analysing how response variability is related to this uncertainty, it also comes with a cost of inconsistency: the Fisher information calculation is based on assumptions of simple Poisson variability (and conditional independence) that are inconsistent with the data (and it is this very systematic inconsistency that is central to the analysis of variability). This should be discussed, and I think the authors will agree that ultimately a (e.g. behavioural) measure of uncertainty that is altogether independent of the analysed neural responses should be used to identify its neural representation.*

Author response: A behavioral measure of uncertainty would indeed be the gold standard here. We refer to our statistic as a proxy in the main text and now mention this idea explicitly in the discussion.

- *p.8, column right, para 2: "V1 neurons whose responses resulted from linear filtering followed by stochastic divisive normalization" → As I understand from the Methods (see also below) you did not actually use your stochastic divisive normalization model to simulate responses, but instead used it simply to compute condition-dependent gain variances, and then used these to simulate responses from the modulated Poisson model (with a gamma distribution for gain that had the given variance). If my understanding is correct then the sentence quoted above is misleading and needs to be refined.*

Author response: We have adapted that sentence to clarify that we are indeed only using the stochastic normalization model to compute the mean firing rate and gain variability.

- Fig.8b, bottom panel, and c: I couldn't find how estimation variance is defined, please define it. As it seems to be measured in deg^2 it suggests that simply sample variance was used for a circular variable. If so, is there any reason why not a proper circular statistic of dispersion (eg mean resultant length) was used?

Author response: Thank you for catching this. We now use circular variance, which is defined as one minus the length of the mean resultant vector. We updated Figure 8 and its legend accordingly.

- page 6, penultimate para: "suggesting that they induce stimulus uncertainty for different reasons" → I didn't follow the logic here.

Author response: We have replaced this sentence with "suggesting that they independently contribute to stimulus uncertainty".

- *ibid.* "This suggests that gain variability represents the total amount of stimulus uncertainty, regardless of the source of this uncertainty" and p.7, end of para 1 "This invariance to the source of uncertainty...", → But there are multiple cases when different conditions in Fig.3b correspond to the same level of uncertainty (y-axis) but have different amounts of gain variability (x-axis). Do these cases contradict these claims? Please discuss.

Author response: Good point. The data are scattered and hence somewhat ambiguous to interpret. We have softened the language to: "This suggests that gain variability may represent the total amount of stimulus uncertainty"

- There are multiple references in the decoding section (and associated methods and Fig.7 caption) to how the "dynamic range" of cells has been randomly chosen — but I couldn't find the (either formal or informal) definition of dynamic range anywhere.

Author response: We added a couple of sentences to the Methods section to clarify this.

- p.8, right column, end of para 1: "consistently with its interpretation as a consequence of stochastic normalization" → Why couldn't stochastic normalization take place on a short time scale in principle? Your specific stochastic normalization model does indeed have a slow time scale — but even that is only because you choose epsilon to be fixed for the duration of a trial.

Author response: We agree with the reviewer that this is a "soft" prediction in that it depends on the dynamics of the normalization signal which are believed to contain a slow time-scale, but are not well characterized empirically. We have removed the "consistently..." sentence.

- p.14, para 1, 1st sentence: Give the functional form of g_i , and then explain further down where you talk about fitting mean responses how you fitted this form to data. And be more explicit about whether you fitted all neurons' data at once with a single set of parameters p and β , or you fitted each neuron's data separately and then averaged the best-fit parameter values across neurons. (I assume the "average estimate" expression implies the latter but better be sure.)

Author response: We estimated the parameters of the uncertainty receptive field using a simple heuristic. We fit each neuron's data separately using the divisive normalization model in (Goris, 2015), and use their population-average in our model. Fixing those parameters we then estimate the remaining (normalization noise) parameter from patterns of population-averaged gain variability. Note that g_i does not enter into the definition of the uncertainty receptive field and thus is not required for fitting the patterns of gain variability. We have clarified the sentences of the Methods which describe this procedure.

- p.14, last para of "Fitting the stochastic normalization model": "The stimulus-dependent normalization $\sum_j g_j(S)$ was computed by simulating responses of a fixed pool of neurons with a diverse set of tuning properties" → Give details about how this was done exactly, i.e. what were the set of $g_j(S)$ simulated. Also, did you assume the same pool for all neurons fitted?

Author response: The normalization pool $\sum_j g_j(S)$ is the same as was used in (Goris, 2015), which was fixed and common across all cells. We have clarified the sentence of the Methods which describes this.

- p.15, para 2: "We then applied the equations of the stochastic normalization model to obtain a firing rate and gain variability for each neuron." → What were the cells of the normalization pool in this case? Were these the same cells that were decoded, i.e. did you assume that you observe a complete population in which neurons divisively normalize one another? If so, shouldn't you look at how robust your decoding results are in the more realistic case if you only observe a subset of the population of neurons that provide divisive normalization to the neurons you are decoding (and when the divisive normalization pools of your decoded neurons only overlap partially)?

Author response: As mentioned above, we assume a fixed, generic normalization pool which is independent from the neurons being fitted. Hence we do not assume that we observe the neurons in the normalization pool.

- p.15, after the first set of equations (btw, you should consider numbering your equations): say explicitly what $f_i(s)$ is in the equation for λ_i (I assume it's the same as in the stochastic normalization model).

Author response: That is indeed the case, we now make this explicit.

- p.15, left column, last para: "with a random initialization" → If this means that you optimised the likelihood starting from a single initial condition, explain why you treated the likelihood as concave — otherwise you would have needed to try several initial conditions.

Author response: We did use several initial conditions, we now make this explicit in the text.

- p.15, interneuronal gain correlations: Once you create G by adding two gamma variates of different scale (because G_s and G_p are multiplied by different constants), I don't think it is gamma distributed any more — yet your inverse binomial decoding likelihood (implicitly) assumes a gamma distributed gain. Please comment on this. I also suggest using a symbol other than lambda for the mixing ratio of shared and private gain variability to avoid confusion with λ_i used elsewhere (including the next paragraph).

Author response: There is indeed a model mismatch due to the distribution of the gain variable, but its influence seems fairly minor given the good performance of the uncertainty decoder with long integration times, even when noise-correlations are high. More important seems to be the overall signal-to-noise ratio as indicated by the larger effect of integration time. We have fixed that typo, replacing λ with γ .

- I would like to encourage you to consider sharing your data more openly than just "upon reasonable request".

Author response: We appreciate the encouragement. We are currently making a website for the Goris-lab. This website will contain a section with downloadable data, including the data-sets used for this paper. Given that this website is not online yet, we will stick to the "upon reasonable request" language in the paper.

- Ref 26 has been published (in *J Neurosci*) in the mean time.

Author response: Fixed.

Máté Lengyel

Reviewer #3 (Remarks to the Author):

This is a comprehensive response and the authors have dealt with all of my initial concerns.

Nicholas Price

Reviewers' Comments:

Reviewer #1:

Remarks to the Author:

Thanks for the new revision. I appreciate the authors' effort to clarify the paper's contribution and I think the text is now much more balanced than in its original version. We still have differences in opinions regarding the other point I raised, the relative importance of the three axes of investigation and where different models are positioned on those axes.

However, I completely agree that beyond a certain point, insisting on further clarifications is counterproductive. In other words, I agree with the authors on the sufficient benefits of "agreeing on points of disagreement" rather than pursuing an ultimate (and likely impossible) agreement in views. Consequently, I have no further comments to address regarding the manuscript. Congratulations on an interesting work!

Jozsef Fiser

Reviewer #2:

Remarks to the Author:

I thank the authors their comprehensive responses and the changes they have made to the manuscript. They have pretty much addressed all my main comments, bar one — unsurprisingly the one concerning the sampling vs PPC dichotomy and where this paper stands on that.

I think we can agree to disagree on many issues but I would like to fix a factual misunderstanding in the authors' response. On p.5 of their response they write: "The model used in Supplementary Figure 5 of Orban et al (2016) is indeed a spatial variant of the sampling hypothesis" and that "However, note that the model presented in the introduction and Figure 1 of Orban et al is explicitly time-based". This is incorrect: **all** figures of that paper (including Figs. 1 and S5) correspond to the **exact same sampling model**. This was precisely the point I was trying to make in my previous review: that a "time-based" sampling model can be very well decoded as a "spatial" (ie. population coding) model when the decoded variable (a single scalar orientation variable) is much lower dimensional than the latent variable space (248 latent variables) from which the model is sampling. (See also Shivkumar et al, NeurIPS 2018 for a related point with analytical results.)

In light of this misunderstanding, and given that the whole "recurrent/iterative" vs "feedforward/instantaneous" distinction in the 3rd paragraph of the Discussion actually rests on whether the population can be decoded in short time windows (as we have clarified that mechanistically, divisive normalization may also require recurrent interactions), I think that the claim that the author's model, and empirical results supporting it, are closer to PPC than sampling along this axis needs to be significantly dampened. In comparison, I find it ironic that the authors have now actually added a disclaimer to their model being closer to sampling than PPCs along the primary axis of variance and mean being vs not being independent information channels — based on **unpublished** results on an undoubtedly interesting but as yet hypothetical variant of PPCs that has not been compared to experimental data systematically (though I do appreciate Xaq Pitkow's very knowledgeable response from the sidelines), while they continue to ignore published results in a peer-reviewed journal (Orban et al, 2016; see above) that directly call into question the relevance of their secondary axis and the associated claim they make.

At this point, I will leave it up to the authors how they deal with this issue in the final version of the paper — I feel that one more iteration would not only be overly onerous to all of us involved (and, no

doubt, primarily to the authors) but, given what has happened in this last revision, might just risk even more softening of the primary axis and boosting of the secondary (-ies) which I really think is the wrong direction of travel here.

More minor comments:

- It is emphasised several times in the text (abstract, para 4 of introduction, last para of "Testing the theory in visual cortex") that the tuning of gain variability is "invariant to the source of uncertainty" — which indeed would be a theoretically important property. On the up side, I see how the results on the example neuron shown in Fig. 2f support this: all points are along the same line to a good approximation, and in particular, points corresponding to different manipulations but resulting in similar amounts of uncertainty — dark purple, light red — correspond to the same amount of gain variability. However, this pattern is much less clear, and in fact seems to be violated by several data points in the whole populations (Fig.3b). [The analysis of "selectivity" in Fig.3e is somewhat misleading in this regard: because the extreme manipulations along the two axes may correspond to very different amounts of uncertainty change, the fact that gain variability is changed by the same amount between them does not support the "same amount of uncertainty (change) = same amount of gain variability (change)" claim suggested by this clause.] Moreover, this (Figs.2f) kind of analysis is not even attempted in relation to Fig.4 — ie. computing Fisher (or perhaps more appropriately in this case: mutual) information about texture in V2 neurons under manipulations of image type (texture vs. noise) and contrast, using it as an (inverse) measure of uncertainty, and again plotting the relation between uncertainty to gain variability across all conditions of both manipulations, and showing the kind of result that you have in Fig. 2f. In lack of such supporting evidence (Figs.3b, 4) I think this claim needs to be softened at all relevant places in the text. You have already agreed (and implemented) to soften this claim at one point in the text in this revision — so hopefully you will agree to bring the other points in sync with it. Eg. in the abstract you don't "show that [...] this tuning is invariant to the source of uncertainty" but your "results suggest that this tuning may be invariant to the source of uncertainty".

- Fano factors: I appreciate the explanation and supplementary figure the authors added to the manuscript to explain why the Fano factor fails as a neural correlate of uncertainty. But it's precisely Supplementary Fig. 1 that revealed what the issue may be with this analysis. First of all, the authors should be more forthcoming about the fact that their results seem somewhat at odds with previous results showing significant modulations of Fano factors by contrast (Orban et al, 2016, analysing data from Ecker et al, 2010 — and also in the limit of no contrast vs high contrast, see Churchland et al, 2010). Second, the main difference is not (as the authors claimed in their response to the previous revision) that these previous results were obtained by mean matching, but by how exactly the population-average Fano factor (FFpop) was computed. In these previous papers it was computed as $FF_{pop} = \langle \sigma^2_i \rangle / \langle \mu_i \rangle$, where μ_i and σ^2_i are the spike count means and variances of neuron i (or neuron-condition pair i), $\langle . \rangle$ denotes averaging across the population (so across i) performed using an uncertainty-weighted average (depending on the uncertainty in the variance estimates). So geometrically, this is the slope of the line fitted between the origin and the center of mass of the scatter plot ("the slope of the [weighted] average"). In contrast, the authors seem to have computed it as $FF_{pop} = \langle \sigma^2_i / \mu_i \rangle$ ("the [unweighted] average of the slopes" of the lines going from the origin to individual points in the scatter plot) which is prone to be more sensitive to changes in overall responsiveness in the way Fig. S1 illustrates. I appreciate that the authors may not want to rerun their analyses with the "original" definition of FFPop at this point, but at the very least they should discuss this issue.

- p.11, right column, 1st para: "The mean response of this model $f(S)$ is identical" → "The mean response of this model $f(S)$ is approximately equal". This is based on a 1st order Taylor expansion.

Btw, the accuracy of this approximation never seems to be tested: shouldn't you simulate the full divisive normalization model (equation following Eq.1) after you have fitted its σ_N parameter to data based on your Taylor expansion-based approximation, and see how well the (numerically computed) mean and std of responses in the full model match the (analytically obtained) mean and std of responses under the approximate model (equations preceding Eq.2)?

- ibid. "yields a simple expression" → "yields a simple approximate expression"

- p.17. Equation for the inverse Fisher information: you define it as $1/I = E[h/h'^2]$, that is $1/I = E[1 / (h'^2/h)]$, but shouldn't it be $1/I = 1/ E[h'^2/h]$ (see e.g. Eq.3.45 in Dayan&Abbott where $I \sim E[h'^2/h]$). Unless I just got something plain wrong (possible), please explain / discuss why you are computing it in your way.

Máté Lengyel

Response to reviewers

We again thank the reviewers for their constructive feedback. We report the reviewers' comment in italic and include our point-by-point responses in regular font. Finally, we have highlighted all changes to the updated manuscript in bold.

Reviewer #1 (Remarks to the Author):

Thanks for the new revision. I appreciate the authors' effort to clarify the paper's contribution and I think the text is now much more balanced than in its original version. We still have differences in opinions regarding the other point I raised, the relative importance of the three axes of investigation and where different models are positioned on those axes. However, I completely agree that beyond a certain point, insisting on further clarifications is counterproductive. In other words, I agree with the authors on the sufficient benefits of "agreeing on points of disagreement" rather than pursuing an ultimate (and likely impossible) agreement in views. Consequently, I have no further comments to address regarding the manuscript. Congratulations on an interesting work! Jozsef Fiser

Author response:

We thank the reviewer for all of his comments, which helped us to improve our paper substantially.

Reviewer #2 (Remarks to the Author):

*I thank the authors their comprehensive responses and the changes they have made to the manuscript. They have pretty much addressed all my main comments, bar one — unsurprisingly the one concerning the sampling vs PPC dichotomy and where this paper stands on that. I think we can agree to disagree on many issues but I would like to fix a factual misunderstanding in the authors' response. On p.5 of their response they write: "The model used in Supplementary Figure 5 of Orban et al (2016) is indeed a spatial variant of the sampling hypothesis" and that "However, note that the model presented in the introduction and Figure 1 of Orban et al is explicitly time-based". This is incorrect: *all* figures of that paper (including Figs. 1 and S5) correspond to the *exact same sampling model*. This was precisely the point I was trying to make in my previous review: that a "time-based" sampling model can be very well decoded as a "spatial" (ie. population coding) model when the decoded variable (a single scalar orientation variable) is much lower dimensional than the latent variable space (248 latent variables) from which the model is sampling. (See also Shivkumar et al, NeurIPS 2018 for a related point with analytical results.) In light of this misunderstanding, and given that the whole "recurrent/iterative" vs "feedforward/instantaneous" distinction in the 3rd paragraph of the Discussion actually rests on whether the population can be decoded in short time windows (as we have clarified that mechanistically, divisive normalization may also require recurrent interactions), I think that the claim that the author's model, and empirical results supporting it, are closer to PPC than sampling along this axis needs to be significantly dampened. In comparison, I find it ironic that the authors have now actually added a disclaimer to their model being closer to sampling than PPCs along the primary axis of variance and mean being vs not being independent information channels — based on *unpublished* results on an undoubtedly interesting but as yet hypothetical variant of PPCs that has not been compared to experimental data systematically (though I do appreciate Xaq Pitkow's very knowledgeable response from the sidelines), while they continue to ignore published results in a peer-reviewed journal (Orban et al, 2016; see above) that directly call into question the relevance of their secondary axis and the associated claim they make. At this point, I will leave it up to the authors how they deal with this issue in the final version of the paper — I feel that one more iteration would not only be overly onerous to all of us involved (and, no doubt, primarily to the authors) but, given what has happened in this last revision, might just risk even more softening of the primary axis and boosting of the secondary (-ies) which I really think is the wrong direction of travel here.*

Author response:

We thank the reviewer for pointing out our misunderstanding of the supplementary figure of the Orban (2016) paper and for clarifying his claim that time-based sampling models under certain circumstances can be decoded as a spatial model. We don't disagree with that statement. But this is not the foundation of our argument. Our logic rests on our finding that gain dynamics are slow, not fast. In our view, that result is incompatible with time-based sampling models which explicitly propose fast firing rate dynamics. It is of course consistent with sampling models that are space-based to begin with. We think that our statement in the 3rd paragraph of the discussion is correct. But we have added a couple sentences to the corresponding results section to clarify our logic.

Reviewer #2 (More minor comments):

- It is emphasised several times in the text (abstract, para 4 of introduction, last para of "Testing the theory in visual cortex") that the tuning of gain variability is "invariant to the source of uncertainty" — which indeed would be a theoretically important property. On the up side, I see how the results on the example neuron shown in Fig. 2f support this: all points are along the same

line to a good approximation, and in particular, points corresponding to different manipulations but resulting in similar amounts of uncertainty — dark purple, light red — correspond to the same amount of gain variability. However, this pattern is much less clear, and in fact seems to be violated by several data points in the whole populations (Fig.3b). [The analysis of "selectivity" in Fig.3e is somewhat misleading in this regard: because the extreme manipulations along the two axes may correspond to very different amounts of uncertainty change, the fact that gain variability is changed by the same amount between them does not support the "same amount of uncertainty (change) = same amount of gain variability (change)" claim suggested by this clause.] Moreover, this (Figs.2f) kind of analysis is not even attempted in relation to Fig.4 — ie. computing Fisher (or perhaps more appropriately in this case: mutual) information about texture in V2 neurons under manipulations of image type (texture vs. noise) and contrast, using it as an (inverse) measure of uncertainty, and again plotting the relation between uncertainty to gain variability across all conditions of both manipulations, and showing the kind of result that you have in Fig. 2f. In lack of such supporting evidence (Figs.3b, 4) I think this claim needs to be softened at all relevant places in the text. You have already agreed (and implemented) to soften this claim at one point in the text in this revision — so hopefully you will agree to bring the other points in sync with it. Eg. in the abstract you don't "show that [...] this tuning is invariant to the source of uncertainty" but your "results suggest that this tuning may be invariant to the source of uncertainty".

Author response:

We agree with the reviewer and have softened the claim at all relevant places in the text.

- Fano factors: I appreciate the explanation and supplementary figure the authors added to explain why the Fano factor fails as a neural correlate of uncertainty. But it's precisely Supplementary Fig. 1 that revealed what the issue may be with this analysis. First of all, the authors should be more forthcoming about the fact that their results seem somewhat at odds with previous results showing significant modulations of Fano factors by contrast (Orban et al, 2016, analysing data from Ecker et al, 2010 — and also in the limit of no contrast vs high contrast, see Churchland et al, 2010). Second, the main difference is not (as the authors claimed in their response to the previous revision) that these previous results were obtained by mean matching, but by how exactly the population-average Fano factor (FFpop) was computed. In these previous papers it was computed as $FF_{pop} = \langle \sigma_i^2 \rangle / \langle \mu_i \rangle$, where μ_i and σ_i^2 are the spike count means and variances of neuron i (or neuron-condition pair i), $\langle \cdot \rangle$ denotes averaging across the population (so across i) performed using an uncertainty-weighted average (depending on the uncertainty in the variance estimates). So geometrically, this is the slope of the line fitted between the origin and the center of mass of the scatter plot ("the slope of the [weighted] average"). In contrast, the authors seem to have computed it as $FF_{pop} = \langle \sigma_i^2 / \mu_i \rangle$ ("the [unweighted] average of the slopes" of the lines going from the origin to individual points in the scatter plot) which is prone to be more sensitive to changes in overall responsiveness in the way Fig. S1 illustrates. I appreciate that the authors may not want to rerun their analyses with the "original" definition of FFPop at this point, but at the very least they should discuss this issue.

Author response:

We have added a sentence to the relevant part of the Methods section that points out that some other studies used a different computation to obtain a single value of Fano factor across conditions.

- p.11, right column, 1st para: "The mean response of this model $f(S)$ is identical" → "The mean response of this model $f(S)$ is approximately equal". This is based on a 1st order Taylor expansion. Btw, the accuracy of this approximation never seems to be tested: shouldn't you simulate the full divisive normalization model (equation following Eq.1) after you have fitted its σ_N parameter to data based on your Taylor expansion-based approximation, and see how well the (numerically computed) mean and std of responses in the full model match the (analytically obtained) mean and std of responses under the approximate model (equations preceding Eq.2)?

Author response:

We have adjusted the language as suggested by the reviewer. We did verify the accuracy of the approximation – it is very reasonable in the range of our parameter estimates. Although we think it is important that we know that the approximation is a good one, we opted not to document this exercise in the paper or in supplementary information.

- p.17. Equation for the inverse Fisher information: you define it as $\frac{1}{I} = E[\frac{h}{h'^2}]$, that is $\frac{1}{I} = E[\frac{1}{(h'^2/h)}]$, but shouldn't it be $\frac{1}{I} = \frac{1}{E[\frac{h'^2}{h}]}$ (see e.g. Eq.3.45 in Dayan and Abbott where $I = E[h'^2/h]$). Unless I just got something plain wrong (possible), please explain / discuss why you are computing it in your way.

Author response:

Thank you for catching this. We computed it the way you describe, but misstated the equation in the previous versions of the manuscript. We have fixed this now.